# Evolution of microscopic heterogeneity and dynamics in choline chloride-based deep eutectic solvents

Stephanie Spittle[1,17], Derrick Poe[2,17], Brian Doherty[3], Charles Kolodziej[4], Luke Heroux[5,6], Md Ashraful Haque[7], Henry Squire[8], Tyler Cosby[9], Yong Zhang[2], Carla Fraenza[10], Sahana Bhattacharyya[10], Madhusudan Tyagi[11,12], Jing Peng[13], Ramez A. Elgammal[1], Thomas Zawodzinski[1], Mark Tuckerman[3,14,15], Steve Greenbaum[10], Burcu Gurkan[8], Clemens Burda[4], Mark Dadmun[7,16], Edward J. Maginn[2✉] & Joshua Sangoro[1✉]

Deep eutectic solvents (DESs) are an emerging class of non-aqueous solvents that are potentially scalable, easy to prepare and functionalize for many applications ranging from biomass processing to energy storage technologies. Predictive understanding of the fundamental correlations between local structure and macroscopic properties is needed to exploit the large design space and tunability of DESs for specific applications. Here, we employ a range of computational and experimental techniques that span length-scales from molecular to macroscopic and timescales from picoseconds to seconds to study the evolution of structure and dynamics in model DESs, namely Glyceline and Ethaline, starting from the parent compounds. We show that systematic addition of choline chloride leads to microscopic heterogeneities that alter the primary structural relaxation in glycerol and ethylene glycol and result in new dynamic modes that are strongly correlated to the macroscopic properties of the DES formed.

[1] Department of Chemical and Biomolecular Engineering, University of Tennessee, Knoxville, TN 37996, USA. [2] Department of Chemical and Biomolecular Engineering, University of Notre Dame, Notre Dame, IN 46556, USA. [3] Department of Chemistry, New York University, New York, NY 10003, USA. [4] Department of Chemistry, Case Western Reserve University, Cleveland, OH 44106, USA. [5] Department of Materials Science and Engineering, University of Tennessee, Knoxville, TN 37996, USA. [6] Oak Ridge National Laboratory, Neutron Sciences Division, Oak Ridge, TN 37830, USA. [7] Department of Chemistry, University of Tennessee, Knoxville, TN 37996, USA. [8] Department of Chemical and Biomolecular Engineering, Case Western Reserve University, Cleveland, OH 44106, USA. [9] School of Mathematics and Sciences, University of Tennessee Southern, Pulaski, TN 44106, USA. [10] Department of Physics and Astronomy, Hunter College, New York, NY 10065, USA. [11] NIST Center for Neutron Research, Gaithersburg, MD 20899, USA. [12] Department of Materials Science and Engineering, University of Maryland, College Park, MD 20742, USA. [13] School of Materials Science and Engineering, Beihang University, Beijing, China. [14] Courant Institute of Mathematical Science, New York University, New York, NY 10012, USA. [15] NYU-ECNU Center for Computational Chemistry at NYU Shanghai, Shanghai, China. [16] Oak Ridge National Laboratory, Chemical Sciences Division, Oak Ridge, TN 37830, USA. [17] These authors contributed equally: Stephanie Spittle, Derrick Poe. ✉email: ed@nd.edu; jsangoro@utk.edu

Deep eutectic solvents (DESs) are an emerging class of materials with exceptional properties that make them promising for applications in solar cells[1], redox flow batteries[2–4], thermoelectric energy conversion[5], supercapacitors[6], chemical sensors[7], chemical separation processes[8], drug solubility[9], and environmentally benign solvents for chemical synthesis and biomass processing[10,11]. DESs are mixtures of two or more species, typically a hydrogen bond donor (HBD) species and a hydrogen bond acceptor (HBA) species, that results in a depressed melting temperature significantly below those of the parent compounds[12–15]. The name deep eutectic solvent was first coined for this new class of solvents by Abbott et al[16]. DESs are often considered a subclass of ionic liquids because of their similar properties such as low volatility, low flammability, and wide electrochemical and thermal stability windows[17–19]. In contrast to standard ionic liquids, these materials are often inexpensive to synthesize, and can usually be made from bio-degradable, and nontoxic constituents[13,20]. Although there are many possible combinations of HBAs and HBDs available to form DESs, predictive understanding of structure–property relationships necessary to design DESs for specific applications remains elusive. For instance, DESs exhibit relatively high viscosities, which limits their uses in many technologies, but there is no generally established approach to tune viscosities based entirely on the knowledge of the HBA and HBD chemistry. Although DES-related publications have now surpassed 3000, considerable work still needs to be done to unravel the fundamental relationships between the molecular structure and the macroscopic properties. Detailed fundamental studies of DESs have been largely limited to a few salt-based DESs usually at a single concentration. The depression in melting temperature is typically attributed to the complexation of the HBA, specifically the ions in salt-containing DESs, by the HBD through hydrogen bonding[15,16,19]. Hydrogen bonded networks (HBN) exist in pure HBDs, and these networks are characterized by unique local and supramolecular dynamics[21–23]. Moreover, in lithium salt and amide-based DESs, Kaur et al. and Subba et al. confirmed the existence of spatial and temporal nanoscale heterogeneity, respectively, which likely results from segregated ionic and HBD domains[24,25]. However, the evolution of such hetero-geneities in DESs with composition as well as their resulting influence on macroscopic transport properties of DESs remain poorly understood.

Many key scientific questions remain unanswered. For instance, (i) how do the local structures and microscopic dynamics evolve with a variation of the composition of the mixtures approaching the eutectic concentration, and (ii) how do spatial and temporal heterogeneities influence the macroscopic properties? To answer these questions, we start with canonical HBDs, namely glycerol and ethylene glycol (EG), and track the evolution of structure and dynamics as the HBA, choline chloride (ChCl), is added using a combination of experimental and computational techniques spanning length-scales from molecular to macroscopic as well as broad ranges of time-scales from picoseconds to seconds. Recent studies of Glyceline by pulsed-field gradient nuclear magnetic resonance spectroscopy (PFG-NMR) revealed that the self diffusivity of choline is lower than glycerol[26]. However, when the same system is probed using quasi-elastic neutron scattering (QENS) over a shorter timescale, choline is found to diffuse faster than glycerol despite the larger size of choline[27]. These studies highlight the need to understand structure and dynamics across multiple time- and length-scales, thereby yielding comprehensive structural and dynamic information necessary to evaluate how microscopic properties, including interactions and local structures, affect the physicochemical properties observed at the macroscopic level.

We seek to understand how the systematic introduction of the HBA alters the structure and dynamics of HBD and dynamic environments of each component and unravel the under-pinnings governing transport properties in DES. To do so, we start with well understood HBDs, glycerol, and EG, and track the evolution of structure and dynamics as the HBA, ChCl, is added up to the eutectic composition (33 mol% ChCl in glycerol and EG, namely Glyceline and Ethaline, respectively) using an array of complementary experimental and computational techniques capable of spanning lengthscales from molecular to macroscopic and timescales from sub-picoseconds to seconds and beyond. Techniques employed include PFG- and proton ($^1$H) NMR, ab initio (AIMD) and classical (CMD) molecular dynamics simulations, QENS and wide-angle neutron scattering (WANS), broadband dielectric spectroscopy (BDS), dynamic mechanical spectroscopy (DMS), femtosecond transient absorption (fs-TA) spectroscopy, and differential scanning calorimetry (DSC). A combination of these methods provides access to a wealth of microscopic information across multiple time- and length scales. For clarity, results on glycerol-based mixtures will be shown in the main article while details of EG systems will be provided in the Supplementary Information. Using $^1$H NMR, WANS, CMD, and AIMD simulations, we observe that as the concentration of ChCl is increased, the chloride ions increasingly interact with glycerol molecules, thereby altering the HBN and weakening the self-interactions of glycerol moieties. In addition, DMS reveals that the HBN of glycerol is substantially weakened but not completely destroyed. The dynamics of choline cations are investigated by QENS, PFG-NMR, and indirectly by BDS, and are shown to exhibit lower mobility than the glycerol molecules. However, as the concentration of ChCl increases towards the eutectic composi-tion, we observe that the mobility of all constituent molecules increases, with choline dynamics becoming faster relative to the mean primary structural relaxation rates. These results together provide detailed insights into the observed increase in glass transition temperature depression, dc ionic conductivity, fluid-ity, and the mean rates of orientational dynamics as ChCl concentration is increased. These fundamental insights form a scientific basis for the understanding of the dominant correla-tions in structure-property relationships of DESs needed for rational design and tunable applications.

## Results

We begin with an investigation of the neat HBD species, glycerol, and then gradually increase the concentration of the HBA species, ChCl, up to the eutectic composition to observe the evolution of the structure and dynamics across multiple time- and length scales.

**Local structure and microscopic interactions.** The microscopic properties are examined first. The $^1$H NMR spectra of the 0–33 mol% ChCl/glycerol mixtures are shown in Fig. 1. These were obtained at 333 K instead of room temperature to improve the spectral resolution. The proton from the hydroxyl group of choline, peak "i", is only observed as a slight shoulder in 5 mol% ChCl due to the low concentration of ChCl. The peak assignment is consistent with previously reported data[28]. The spectra show rather broad peak widths ranging from 6 to 51 Hz full width at half maximum, which is consistent with the high viscosities of the samples (Supplementary Table 1). In general, as the ChCl con-centration increases, the peak width decreases with the exception of peak "f", which may indicate more restricted motion of the methyl group on ChCl, and peak "i" the OH group on ChCl due to exchange processes.

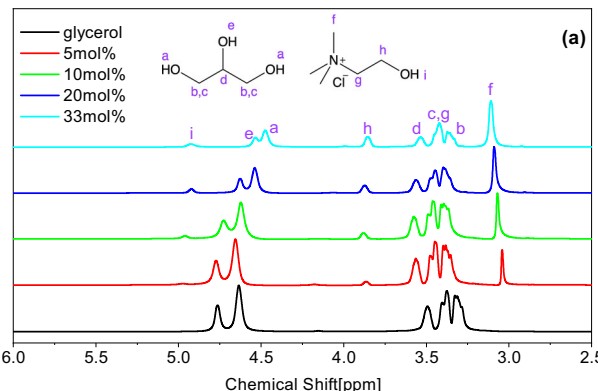
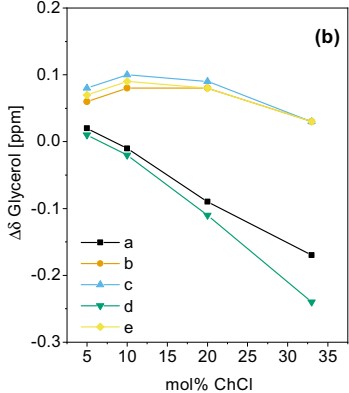

**Fig. 1 Proton NMR spectra and chemical shift changes in ChCl and glycerol mixtures. a** [1]H NMR spectra of glycerol and mixtures of 5, 10, 20, and 33 mol % ChCl in glycerol at 333 K. Purple letters a–f denote the various protons on glycerol (left chemical structure) and ChCl (right chemical structure). **b** Change in chemical shift in ppm with respect to neat glycerol as a function of ChCl concentration. Letters a–e denote the protons as specified in purple in part (**a**).

Interestingly, a pronounced difference in the chemical shift of the hydroxyl protons was observed, in particular for the hydroxyl protons, denoted by a, e, and i (Fig. 1b and Supplementary Table 2). This upfield shift suggests that as the concentration of ChCl is increased, the chloride anions interact with these protons, providing slight increases in electron density. The same trends are observed between neat EG and 33 mol% ChCl in EG, shown in Supplementary Fig. 1. However, the shifts are more pronounced, suggesting that the interactions are stronger. The decrease in chemical shift of the ChCl OH proton as the concentration of ChCl increases also suggests that the intermolecular self hydrogen bonding decreases which contribute to the decrease in $Tg$. Chemical shift differences in this system may be primarily attributed to changes in the hydrogen bonding network, structural changes, and anisotropies, as has been seen in ionic liquids and other DESs[29,30].

To further monitor the evolution of the local structure with a concentration of ChCl, we employ CMD and AIMD simulations to obtain the RDFs and coordination numbers ($N_{coord}$) associated with specific interactions. The CMD force field bulk behavior was first validated by comparing against experimental results. The computed density as a function of composition matched the absolute experimental values quite well, though the simulations predict a slightly greater composition dependence of the density than is observed experimentally (Supplementary Fig. 2). Although the computed viscosity is lower than the experimental viscosity, the trend with composition is captured very well (Supplementary Fig. 3) and should yield reasonable predictions of composition-dependent dynamics. Figure 2a shows the ChCl and glycerol molecules, with examined atoms and interactions highlighted. Center and terminal hydrogen and oxygen atoms of glycerol are referred to as Hc, Ht, Oc, and Ot, respectively. The hydroxyl group on choline is labeled Hy and Oy for the hydrogen and oxygen atoms respectively. Figure 2b shows the comparison of an overall RDF for the 33 mol% ChCl system from AIMD and CMD, where peak shapes and positions are in excellent agreement with each other.

Partial RDFs are also provided (Supplementary Fig. 4), where pair-wise interactions with chloride are compared between AIMD and CMD. A small shift in the peak heights and distances is observed, however, this result has been seen in a previous AIMD study for a 1:1 ratio of ChCl to glycerol by Korotkevich et al.[31], and is likely due to the inclusion of polarization effects in the AIMD approach. In addition, differences in peak heights can be due to many complicating factors such as system size, AIMD accounting for many-body terms, and the high sensitivity of

RDFs to these influences. A more reliable measurement of local solvation structure is through $N_{coord}$. Integration of the RDF over the first solvation shell, defined as the first minimum after the first maximum in the RDF, provides $N_{coord}$ values that indicate the general local structuring. Table 1 summarizes the $N_{coord}$ corresponding to the local solvation environment with respect to the first atom in the site-site column. The results show a good agreement for $N_{coord}$ between CMD and AIMD. Local structuring was also compared with the calculated neutron scattering function $S(q)$ at 400 K for both AIMD and CMD and provided in Supplementary Fig. 5. The plots are nearly identical at this temperature, indicating that the force field is able to capture the relevant structural features.

Figure 2c provides an additional comparison of CDFs with an RDF on the $x$-axis and an angular distribution function on the $y$-axis created from AIMD and CMD simulations at the eutectic composition, 33 mol% ChCl in glycerol. Assessments of the HBN of DESs have been performed in the literature, where a maximum donor–acceptor distance has been set to either 3.0[32] or 3.5 Å[33] with a minimum O–H–X angle of 150°[33] or 130°[32]. The high-intensity regions in Fig. 2c are consistent with even the most stringent hydrogen bonding criteria. In AIMD, chloride has an average donor-acceptor distance of 2.1 Å and a high probability of bonding angles greater than 150°. In CMD, the average donor–acceptor distance is 2.4 Å and a high probability of bonding angles greater than 135°. Based on the above discussion, agreement across the overall RDF, $N_{coord}$, S(q), and CDFs suggests that the force field used in the CMD simulations adequately captures the local structure of Glyceline, and can thus be used for large scale and long simulation times needed for analysis of dynamics.

The $N_{coord}$ shown in Table 1 provides quantitative insight about the hydrogen-bonded nature of 5 and 33 mol% ChCl in glycerol. AIMD simulations at 33 mol% indicate that the chloride anion does not selectively coordinate with the Hc or Ht hydrogens of glycerol, with $N_{coord}$ of 0.62 (Hc) and 1.27 (Ht), respectively, considering the 1:2 ratio of Hc:Ht in glycerol. CMD results for 300, 340 (provided in Supplementary Table 4), and 400 K show a slight preference for Ht with a ratio closer to 1:2.5, though this still indicates no strong selectivity of the Cl. These results agree with the [1]H NMR data at 333 K in Fig. 1, which shows a uniform shift of the Hc (e) and Ht (a) peaks as ChCl concentration is increased, rather than heterogeneous shift between the two protons that would indicate preferential association with chloride. At 300 K, however, the picture is slightly different, where Ht is moderately favored over Hc. The ratio of Hc and Ht coordinating with Cl

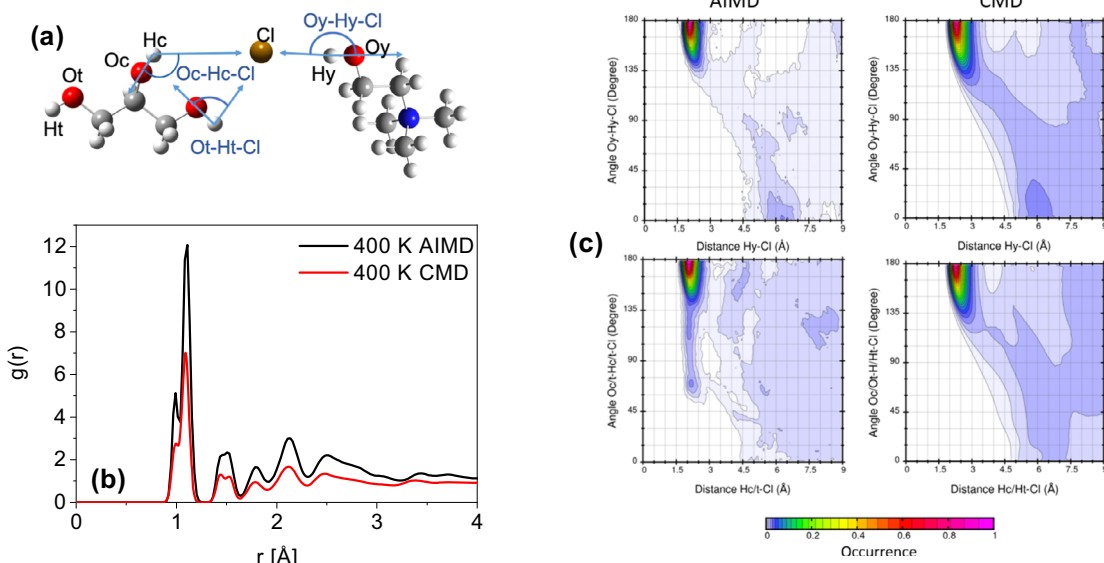

**Fig. 2 Simulated RDFs and CDFs by AIMD and CMD showing reasonable quantitative agreement. a** The labeled atoms in black, Ht, Ot, Oc, Hc, Oy, Hy, and Cl, represent the different atoms in glycerol (left) and ChCl (right). The angles between atoms defined for the combined distribution functions are expressed in blue. **b** Comparison of CMD and AIMD overall radial distribution functions of 33 mol% ChCl in glycerol at 400 K. **c** Combined distribution functions (radial distribution between a hydroxyl hydrogen-Cl on the x-axis and angular distribution, defined in blue text in (**a**), on the y-axis) of chloride to the choline hydroxyl (top) and all three glycerol hydroxyl groups (bottom) from AIMD and CMD simulations at 400 K. Data were normalized to a range between 0 and 1 to generate the color scale provided.

**Table 1 Coordination numbers ($N_{coord}$) for 5 and 33 mol% ChCl in glycerol from CMD and AIMD at the specified temperatures (Gly = glycerol).**

| | | CMD 5 mol% (300 K) | | | CMD 33 mol% (300 K) | | | CMD 33 mol% (400 K) | | | AIMD 33 mol% (400 K) | | |
|---|---|---|---|---|---|---|---|---|---|---|---|---|---|
| Site-site | | $r_{max}$ | $r_{min}$ | $N_{coord}$ | $r_{max}$ | $r_{min}$ | $N_{coord}$ | $r_{max}$ | $r_{min}$ | $N_{coord}$ | $r_{max}$ | $r_{min}$ | $N_{coord}$ |
| Gly–Gly | Oc–Hc | 1.9 | 2.5 | 0.28 | 1.9 | 2.5 | 0.19 | 1.9 | 2.6 | 0.14 | 1.8 | 2.6 | 0.19 |
| | Oc–Ht | 1.8 | 2.5 | 0.50 | 1.8 | 2.5 | 0.31 | 1.9 | 2.5 | 0.26 | 1.8 | 2.5 | 0.39 |
| | Ot–Hc | 1.8 | 2.5 | 0.31 | 1.8 | 2.6 | 0.22 | 1.9 | 2.6 | 0.17 | 1.8 | 2.5 | 0.22 |
| | Ot–Ht | 1.8 | 2.5 | 0.65 | 1.8 | 2.5 | 0.42 | 1.8 | 2.6 | 0.35 | 1.8 | 2.6 | 0.26 |
| Cl–Gly | Cl–Hc | 2.4 | 3.3 | 0.81 | 2.4 | 3.4 | 0.55 | 2.4 | 3.6 | 0.63 | 2.1 | 3.1 | 0.62 |
| | Cl–Ht | 2.3 | 3.3 | 2.33 | 2.3 | 3.4 | 1.42 | 2.3 | 3.6 | 1.54 | 2.1 | 3.2 | 1.27 |
| Cl–Ch | Cl–Hy | 2.4 | 3.4 | 0.11 | 2.4 | 3.5 | 0.50 | 2.4 | 3.6 | 0.52 | 2.1 | 3.0 | 0.62 |
| | Cl–N | 4.6 | 7.3 | 1.09 | 4.0 | 5.3 | 2.53 | 4.3 | 6.4 | 2.95 | 4.4 | 6.7 | 3.72 |
| Gly–Ch | Oc–Hy | 1.9 | 2.5 | 0.01 | 1.9 | 2.6 | 0.08 | 2.0 | 2.6 | 0.08 | 1.8 | 2.5 | 0.09 |
| | Ot–Hy | 1.9 | 2.6 | 0.02 | 1.9 | 2.6 | 0.10 | 1.9 | 2.7 | 0.09 | 1.9 | 2.5 | 0.04 |
| | Oy–Hc | 2.1 | 2.5 | 0.09 | 2.1 | 2.5 | 0.05 | 2.2 | 2.4 | 0.04 | 1.9 | 2.5 | 0.04 |
| | Oy–Ht | 2.0 | 2.4 | 0.11 | 2.1 | 2.4 | 0.05 | 2.2 | 2.3 | 0.05 | 1.9 | 2.5 | 0.15 |
| Ch–Ch | Oy–Hy | 2.1 | 2.3 | 0.00 | 2.1 | 2.3 | 0.01 | 2.1 | 2.3 | 0.01 | 1.9 | 2.6 | 0.04 |

Provided are the peak maxima ($r_{max}$) and peak minima ($r_{min}$) of the first solvation shell determined from radial distribution functions in angstrom. The first listed atom is the reference molecule and the second atom is the observed molecule. CMD coordination numbers for all concentrations are provided in Supplementary Table 3.

remains the same at both low and high HBA concentrations. Choline self-interactions (Oy–Hy) are virtually non-existent at all concentrations. At low ChCl concentrations, choline is primarily solvated by the abundant glycerol molecules. As ChCl concentration is increased, reduced coordination of the oxygen atom on choline by glycerol is observed through a decrease in $N_{coord}$ of Oy–Hc and Oy–Ht. Furthermore, the hydroxyl proton of choline is still mildly associated with glycerol at higher concentrations, indicated by slightly higher Oc–Hy and Oc–Hy numbers. However, there is a much stronger affinity to the chloride anions shown by the significantly elevated Cl–Hy and Cl–N coordination numbers. Chloride functions as an HBA and "linker" that is capable of forming multiple hydrogen bonds with glycerol, as

supported by high coordination numbers of Cl–Hc/t at 33 mol% and much lower Oc/t–Hc/t interactions. On the other hand, all the $N_{coord}$ associated with all Oy are near zero at 300 K, showing that choline does not function as an HBA in any appreciable capacity at room temperature.

For a further understanding of the impact of the HBA concentration on the evolution of the glycerol structure, we performed WANS experiments at 300 K. The structure factors, $S(q)$, of 5 and 33 mol% d-ChCl in d-glycerol are shown in Fig. 3a. The calculated results from CMD simulations at 0, 5, and 33 mol% ChCl are also provided. It is evident that the simulations are able to capture the key features and trends of the experimental $S(q)$ profiles. The peak at approximately 1.5 Å$^{-1}$, likely associated with the first

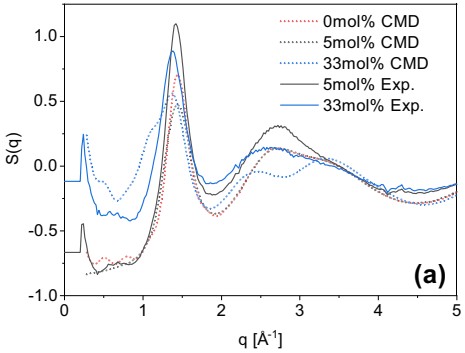
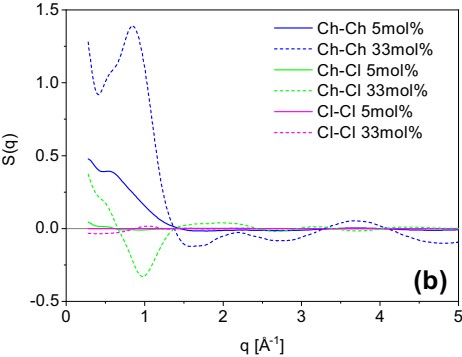

**Fig. 3 Experimental and simulated structure factors for 5 and 33 mol% ChCl in glycerol decomposed. a** Structure factors obtained from CMD simulations (0, 5, and 33 mol% ChCl in glycerol) and WANS (5 and 33 mol% ChCl) at 300 K. **b** Comparison of 5 and 33 mol% for choline–choline, choline–chloride, and chloride–chloride interactions from CMD simulations.

solvation shell of glycerol–glycerol interactions, shifts to longer length scales (lower $q$) with increasing ChCl concentration. This indicates that glycerol self-interactions are weakened as ChCl is introduced, which is consistent with $N_{coord}$ trends. There are some noted deviations between simulations and experiments, seen as a shoulder at $1\,\text{Å}^{-1}$ and peak splitting between 2 and $4\,\text{Å}^{-1}$ in simulated 33 mol%. This is attributed to Ch–Ch interactions and is due to the mixed deuteration state of the experimental choline, something that was seen in Ethaline structure factors of a previous work[34]. A key advantage of the simulated structure factors is the ability to decompose the total profile and isolate individual molecule–molecule and atom–atom interactions. The decomposed Ch–Ch, Ch–Cl, and Cl–Cl correlations are shown in Fig. 3b. At approximately $1.0\,\text{Å}^{-1}$, both Ch–Ch and Cl–Cl correlations show a positive feature whereas Ch–Cl has a negative feature in the 33 mol% ChCl mixture in glycerol, which is the signature of a charge alternation structure[35,36]. In order to rule out ambiguities from mixed deuteration and protiation states, as these have mixed negative and positive form factors, fully deuterated results are provided in Supplementary Fig. 6. These results also exhibit charge alternation occurring at the same $q$ as Fig. 3b.

**Rotational and translational dynamics**. Now, we investigate the evolution of the macroscopic properties of the ChCl and glycerol mixtures to determine how the microscopic changes in interactions of the HBN affect the bulk properties. To do so, BDS, DMS, and fs-TA spectroscopy are employed to obtain information about the dynamics of the mixtures up to the eutectic composition. In Fig. 4a, the derivative representation of the dielectric loss, discussed in Section "Experimental and computational details", is shown for glycerol (top) and 5 mol% ChCl in glycerol (bottom). For dipolar liquids, the primary dielectric relaxation is typically due to reorientations of molecular dipoles and is easily visible as a peak in the dielectric loss. This is commonly referred to as the structural, $\alpha$-relaxation and is associated with the dynamic glass transition[37,38]. In neat glycerol, only the $\alpha$-relaxation is clearly observed. Upon the addition of ChCl, contributions from ion dynamics (dash-dot line) are observed, and a unique, slower relaxation emerges (shaded region). BDS data at a representative temperature for each concentration are provided in Supplementary Figs. 7–11 and all fitting parameters are provided in Supplementary Tables 5–9. At low temperatures, we examine how dynamics change as the glass transition is approached. Additionally, a room temperature BDS measurement of 33 mol% ChCl in glycerol is included in Supplementary Fig. 12 to show that trends and observations in the low-temperature data hold true for higher temperatures as well. The slow, sub-$\alpha$ dielectric relaxation has not been reported in DESs before, presumably because the

data were not analyzed in such detail. Faraone et al. studied BDS of Glyceline and only reported the structural relaxation and ion dynamics, however, the derivative representation of the dielectric loss was not examined, which is where the sub-$\alpha$ relaxation becomes obvious[39]. If considered, the sub-$\alpha$ relaxation is observed in their data as well (Supplementary Fig. 13).

Figure 4b shows a normalized plot of the dynamic mechanical spectra at different concentrations. The curves are normalized by the corresponding zero-shear viscosity ($\eta_0$) and mechanical structural relaxation rate ($\omega_\alpha$) for each concentration for a straightforward comparison of the spectral shapes to evaluate the possible emergence of dynamic heterogeneity in the mixtures. Two distinct dynamic modes are observed for all compositions, namely mechanical structural relaxation and slow mechanical relaxation. This slow mechanical relaxation has been previously attributed to the motion of the HBN of glycerol[40]. It is clear from Fig. 4b that the spectral shapes remain unaltered with the addition of ChCl. This implies that although the mean rates of dynamics are increasing with composition towards the eutectic composition, the general form of the glycerol HBN is largely preserved up to the eutectic composition. The NMR data combined with WANS and MD simulations show that the addition of ChCl does alter the hydrogen bonds of glycerol and the nature of its HBN, however, the dynamic mechanical spectra suggest that the network is not completely disrupted. This is supported by previous reports, stating the HBN of glycerol is largely preserved at 33 mol% ChCl[39,41,42]. Figure 4c, d shows the relaxation dynamics of betaine-30 (Reichardt's) dye in 33 mol% ChCl in glycerol. The absorption maxima and relaxation dynamics of betaine-30 are strongly influenced by its solvent environment. The longer time constant, $\tau_2$, from fs-TA corresponds to orientational dynamics. By comparing the 10 mol% (Supplementary Fig. 14) and 33 mol% ChCl solvent systems, it is evident that the absorption maxima of our probe (betaine-30) remain unchanged between both DES compositions, indicating a similar dielectric constant from the perspective of the probe dye. However, the dynamics at higher ChCl concentrations become faster, and the characteristic timescales obtained are in quantitative agreement with those of structural relaxation measured by BDS.

The mean structural relaxation rates ($\omega_\alpha$) obtained from all the techniques presented in Fig. 4 are shown in Fig. 5a for all mixtures investigated as a function of inverse temperature. The data for each concentration are well described by the Vogel–Fulcher–Tammann equation (Supplementary Eq. 1)[37]. As the temperature is decreased, the rate of dynamics for all relaxation processes is slowed down. In addition, as the concentration of ChCl is increased up to the eutectic composition, the mean rates of the structural relaxation

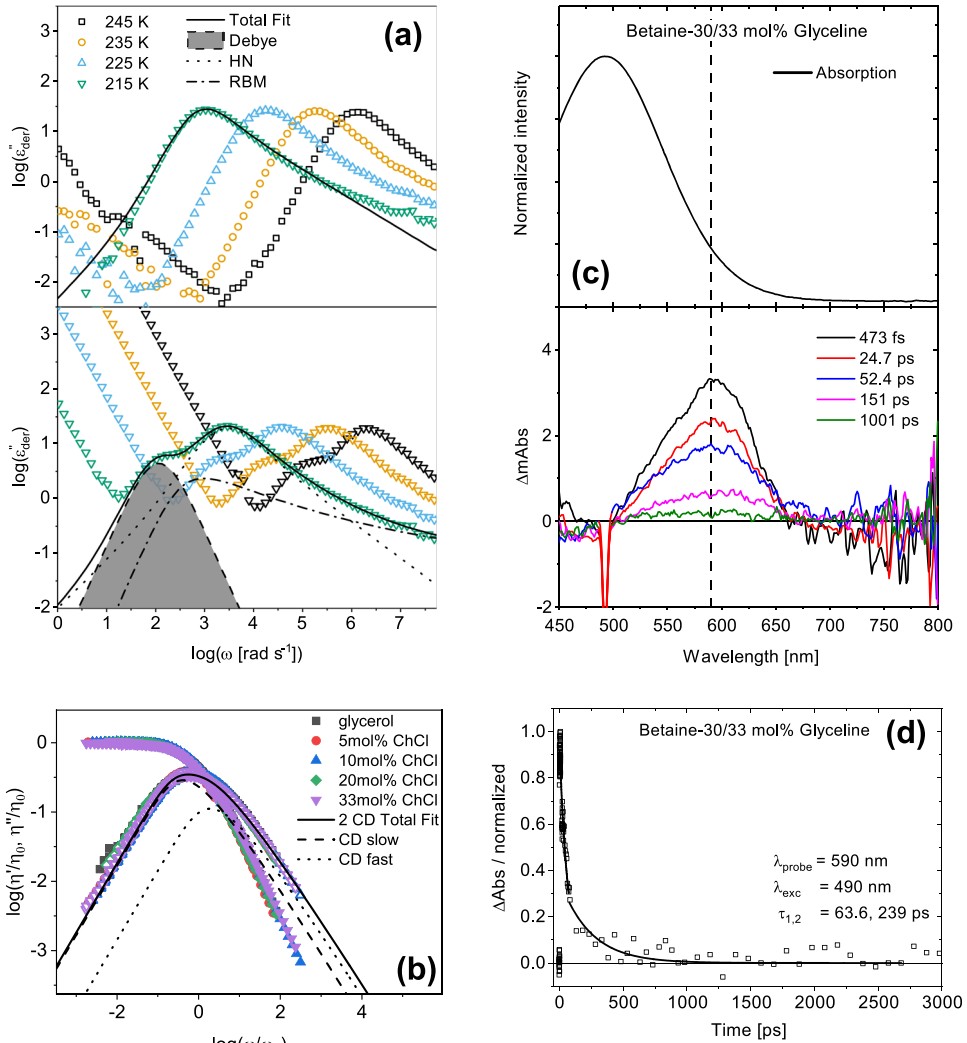

**Fig. 4 Dynamics measured from BDS, DMS, and fs-TA spectroscopy. a** Top: The derivative representation of the dielectric loss, $\varepsilon''_{der}[= -\pi/2\, \partial\varepsilon'/\partial ln(\omega)]$, for glycerol at selected temperatures plotted as a function of radial frequency. Bottom: $\varepsilon''_{der}$ of 5 mol% ChCl in glycerol plotted versus radial frequency at the same temperatures shown for glycerol. The solid black lines denote fits that account for the cumulative contributions observed in the mixtures. For glycerol, the fit comprises of a single empirical Havriliak–Negami function, while the fit for 5 mol% ChCl (and all high ChCl concentrations measured) is the linear addition of a Debye function (shaded region), Havriliak–Negami (dotted line) and Random-Barrier Model (dash-dot line). Fits are described in the SI. **b** The real and imaginary parts of viscosity, $\eta'$ and $\eta''$, for 0, 5, 10, 20, and 33 mol% ChCl in glycerol obtained from a time–temperature superposition as described in the Section "DMS methods", normalized by the corresponding zero-shear viscosity plotted versus angular frequency normalized by the mechanical structural relaxation rate at each concentration. The dotted line represents the probe wavelength at 590 nm. **c** Top panel: Steady-state UV–visible absorption of betaine-30, bottom panel: femtosecond transient absorption spectra at indicated delay times. **d** fs-TA kinetics of betaine-30 in 33 mol% Glyceline at a probe wavelength of 590 nm. The solid line at 0.0 represents the baseline.

become faster. This shows that molecular mobility is increasing, which is consistent with the expectation of lowered melting temperature for the DES studied here[15]. Interestingly, the mean rates of orientational dynamics obtained from the fs-TA, BDS, and DMS are all in quantitative agreement. This remarkable agreement lends credence to our assignment of the primary dielectric relaxation to reorientational dynamics. Figure 5b displays the $\varepsilon''_{der}$ data for the various concentrations plotted versus radial frequency normalized by the corresponding $\omega_\alpha$ of each mixture. This shows the progression of the slow, sub-$\alpha$ relaxation with composition. It is clear that the slow, sub-$\alpha$ relaxation becomes faster relative to the structural relaxation with increasing concentration of ChCl. The same trend is observed for ChCl/EG mixtures, as shown in Fig. 5d. In addition, Fig. 5c shows that the sub-$\alpha$ relaxation rate ($\omega_{slow}$) is coupled to the ion hopping rate ($\omega_{ion}$), and both are decoupled from $\omega_\alpha$ in ChCl/glycerol mixtures. This suggests that the sub-$\alpha$ dielectric

relaxation originates from ion rearrangements. These results are in contrast with recent reports by Faraone et al. and Reuter et al., which suggested a strong coupling between ion dynamics and the structural relaxation[39,43]. Again, their raw dielectric data for Glyceline agrees with ours, although electrode polarization appears at higher frequencies obscuring the slow mode that is revealed using the derivative formalism (Supplementary Fig. 13).

To further understand the potential microscopic origin of the sub-$\alpha$ relaxation observed in BDS, CMD simulations were used to obtain dipole moment rotational correlation functions of choline and glycerol at different compositions. These correlation functions were fit using the fraction kinetic model shown in Eq. (5) (found in the Section "CMD methods"). The $\tau_1$ is attributed to the reoriential motion of the molecules, while the shorter time process $\tau_2$ is assigned to the dynamics associated with hydrogen bond formation and breaking events. All values calculated are

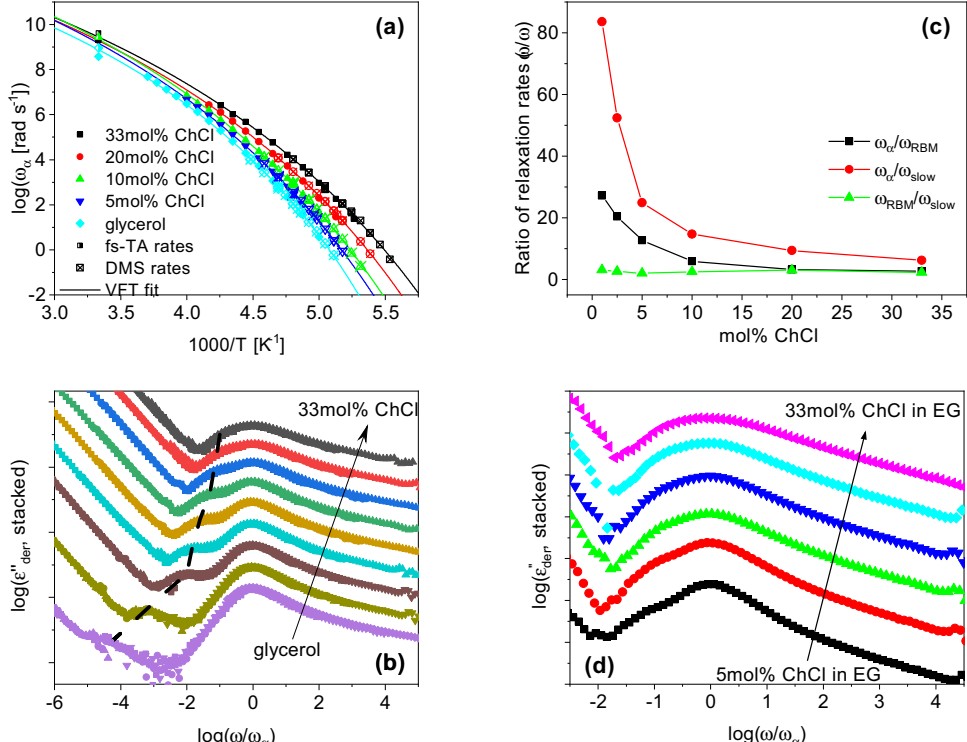

**Fig. 5 The dynamics become faster with increasing concentration. a** Structural relaxation rates from dielectric, mechanical, and transient absorbance spectroscopy are plotted as a function of inverse temperature. The solid lines are fits obtained by the Vogel–Fulcher–Tammann equation. Parameters from these fits can be found in Supplementary Table 10. **b** The $\varepsilon''_{der}$ data for concentrations of 0, 0.05, 0.5, 1, 2.5, 5, 10, 20, and 33 mol% ChCl in glycerol plotted versus radial frequency. The data are arbitrarily stacked to clearly show the evolution of the sub-$\alpha$ relaxation with increasing ChCl concentration. The dashed black line is a guide for the eyes to draw attention to the evolution of the sub-$\alpha$ relaxation. **c** The ratios of the various characteristic rates obtained from the dielectric spectra plotted versus mol% ChCl, at a constant $Tg/T$, showing that the slow relaxation is coupled to ion dynamics. **d** The $\varepsilon''_{der}$ data for concentrations of 5, 10, 15, 20, 25, and 33 mol% ChCl in EG plotted versus radial frequency. The data are arbitrarily stacked to clearly show the evolution of the sub-$\alpha$ relaxation with increasing ChCl concentration.

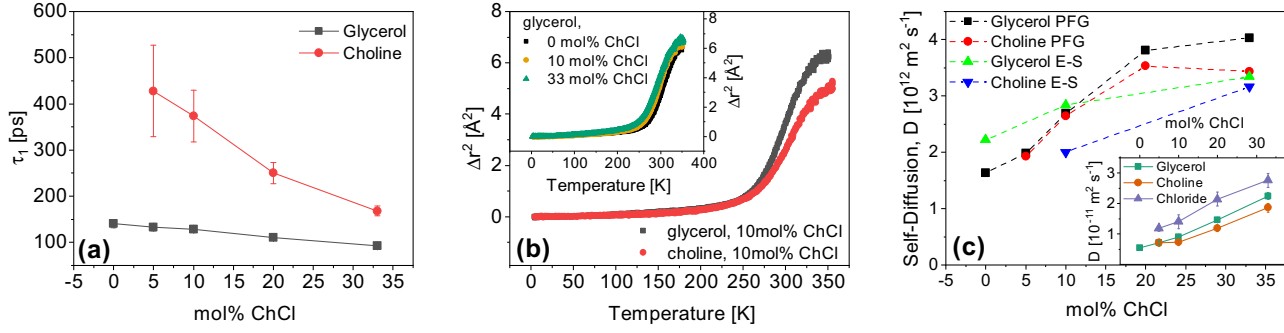

**Fig. 6 Glycerol diffuses and rotates faster than choline at all compositions. a** The average slow characteristic time coefficient ($\tau_1$) of dipole relaxation obtained from CMD simulations for choline and glycerol at various ChCl concentrations in glycerol. Error bars are one standard deviation from five independent simulations and fits. **b** Mean square displacement, $\langle \Delta r^2 \rangle$, of molecules in a mixture of 10 mol% ChCl in glycerol as determined by quasielastic neutron scattering. Inset: MSD of ChCl/glycerol mixtures (black—pure glycerol; orange—10 mol% ChCl; green—33 mol% ChCl. **c** Diffusion coefficients determined from PFG-NMR and calculated from QENS data through the Einstein–Smoluchowski equation, denoted by PFG and E–S, respectively, of glycerol and choline at 298 K for compositions 0, 5, 10, 25, and 33 mol% ChCl in glycerol. Inset: Average self-diffusion coefficients determined from CMD simulations of glycerol, choline, and chloride at 300 K for compositions 0, 5, 10, 20, and 33 mol% ChCl in glycerol. Error bars are one standard deviation from five independent simulations.

shown in Supplementary Table 11. Figure 6a plots $\tau_1$ for choline and glycerol respectively, which shows that glycerol relaxes faster than choline for all mixtures up to the eutectic composition. Based on careful examination of the simulated dynamic profiles, we attribute the slow rotation of the choline cation at low concentrations to the "cage" created by neighboring glycerol molecules that remain interconnected through hydrogen bonding,

leading to spatial and dynamic heterogeneities when viewed at the macroscopic length-scale. Moreover, as ChCl concentration is increased, the glycerol HBN is further weakened, which allows choline, along with the other molecules, to rotate more freely and easily. Choline rotates faster at a fixed timescale of glycerol motion, which is the same trend observed for the sub-$\alpha$ and $\alpha$ dielectric processes, respectively. In addition, considering the

coupling of the sub-$\alpha$ relaxation to ion dynamics shown in Fig. 5c, these results indicate that the origin of the sub-$\alpha$ relaxation is the slow choline rotational dynamics. While the overall dynamic trends observed in mixtures of EG and ChCl by CMD simulations are opposite to those of ChCl and glycerol, data shown in Supplementary Table 12 indicates that choline dynamics are still slower than EG molecules at all concentrations.

Figure 6b shows the mean square displacement (MSD) of the molecules in ChCl/glycerol mixtures over a timescale of approximately 2 ns obtained from QENS experiments. By selective deuteration, the motion of only one component in the mixture is monitored. The inset shows that the addition of ChCl leads to speeding up the motion of the glycerol as the concentration is raised toward the eutectic composition. This is consistent with all the results from other techniques in the current work, further demonstrating that the presence of ChCl modifies the HBN of glycerol, leading to enhanced molecular mobility. Figure 6b documents the motion of the glycerol (black curve) and the choline (red curve) in the 10 mol% ChCl mixture. These data show that the motion of choline is slower than that of glycerol at this concentration. It is interesting to note that the QENS study by Wagle et al. uses a shorter timescale of 0.4 ns, and observes that choline is the faster component[27].

To examine longer-range translational dynamics, self-diffusion coefficients, $D$, were obtained from PFG-NMR and CMD simulations and are shown in Fig. 6c. In addition, the Einstein–Smoluchowski equation was used to estimate the diffusion coefficient from QENS data through $D = \langle \Delta r^2 \rangle / (6t)$, where $t$ denotes the timescale[44]. Although the time- and length-scales of the two experiments differ by nearly three orders of magnitude, the values calculated from QENS agree remarkably with those from PFG-NMR, as shown in Fig. 6c. While CMD dynamics are noticeably faster than PFG-NMR results, this is likely an effect of the significant charge scaling. resulting in faster bulk dynamics as mentioned before. Both methods show that the self-diffusivity of choline is slower than that of glycerol. In general, all the results indicate that the diffusivity increases with the concentration of ChCl. In addition, it is interesting to note from the inset of Fig. 6c that chloride is the fastest species in the mixtures. Locally, CMD dipole moment rotational correlation time coefficients and QENS show that choline becomes faster relative to a fixed glycerol dynamic timescale with increasing ChCl concentration, which is the same trend observed in BDS between the sub-$\alpha$ and structural $\alpha$-relaxation, respectively.

To have a better understanding of the transport properties, the microscopic hydrogen bond dynamics were studied based on CMD simulations. Analysis of hydrogen bond dynamics performed through TRAVIS is based on reactive dynamic flux as laid out by Gehrke et al.[45], with hydrogen bond criteria set to an HBA–HBD maximum angle deviation of 30° and a distance cutoff of the first minima for that interaction as listed in Table 1. This method allows for an understanding of timescales for both a forward and backward movement along a defined reaction coordinate. Forward movement is defined as hydrogen bond breaking and backward movement as hydrogen bond reformation with respective timescales of $\tau_f$ and $\tau_b$. Forward movement can also be understood as the average hydrogen bond lifetime $\tau_{hb}$, making $\tau_f$ and $\tau_{hb}$ equivalent in this definition. With these definitions in mind, the resultant forward and backward time constants for relevant HBA–HBD pairs are listed in Table 2. There is a reduction in both $\tau_f$ and $\tau_b$ in all pair-wise interactions with increasing ChCl save for Oy–Hy interactions that are not altered from the already low $\tau$ values (when considering the standard deviations provided in Supplementary Table 13), indicating a marked increase in overall system fluidity with the addition of ChCl to glycerol, consistent with the trend in the

rotational and translational dynamics. It is important to note the decrease in both the forward and backward time constants, as this implies an overall reduction in the activation energy of hydrogen bond dynamics rather than favoring only hydrogen bond breaking or reformation. It has been reported that the DES Ethaline has a higher viscosity than the corresponding pure HBD, EG[46,47], an opposite trend seen in Glyceline. In a recent study by Zhang et al.[34], it was found that the hydrogen bonds between Cl and EG are much stronger than those between EG molecules and it was believed that these stronger Cl–EG interactions cause the increase in viscosity upon the addition of ChCl to EG. This is supported by [1]H NMR data in Figure S13 as discussed earlier. The results in Table 2 show that, at low ChCl concentrations, the hydrogen bond lifetimes (or $\tau_f$) for Cl–Hc are close to those of Oc–Hc and less than those of Ot–Ht. At high concentrations, Cl–Hc hydrogen bond lifetimes are less than those of both Oc–Hc, and Ot–Hc. The hydrogen bond lifetime of Cl–Ht is slightly longer than those of Oc–Ht and Ot–Ht. These results suggest that the hydrogen bond interactions between Cl and glycerol are weaker or comparable to those between glycerol molecules.

**Macroscopic transport properties and the emerging picture**. The increase in molecular mobility with the addition of ChCl noted in Fig. 5a is consistent with the DSC data shown in Fig. 7a, b, where a decrease in the glass transition temperature ($T_g$) is observed with increasing ChCl concentration. No melting or crystallization is observed in the DSC data, as all of these mixtures are glass-forming liquids. However, if Lindeman's criterion for glass formers is valid such that $T_g \cong 2/3 T_m$[48], the melting and glass transition temperatures are expected to follow a similar trend. Increases in dc ionic conductivity and fluidity are also observed, as shown in Fig. 7c, d. The dc ionic conductivity can be estimated by $\sigma_0 = \Sigma q_i \mu_i n_i$, where $q$ is the charge, $\mu$ is the mobility, and $n$ is the number density of charge carriers[49]. To a good approximation, the mobility of the ions is dominated by the structural dynamics of the matrix on which they are situated. Thus, faster structural relaxation rates would also imply a higher ionic conductivity. The mean structural relaxation rate is also directly related to fluidity by $\omega_\alpha \cong G_\infty \eta^{-1}$, where $G_\infty$ is the high-frequency shear modulus[50]. This relationship qualitatively holds true since as ChCl concentration is increased, the mean relaxation rates and fluidity both increase. While these functional transport relationships hold at a qualitative level, these mixtures do not strictly obey the Stokes–Einstein equation (Supplementary Eq. 2), which is shown in Supplementary Fig. 15. This is not surprising since the moieties and local structures dominating the mass and charge transport are different.

Figure 8a displays a snapshot of 5 mol% ChCl in glycerol, which shows that at this concentration the choline cations are "trapped" by the glycerol network. This result is captured in BDS, CMD simulations, and QENS which all indicate that the dynamics of choline are significantly slower than glycerol at the low ChCl concentrations. Depicted in Fig. 8b, a higher concentration of ions dispersed throughout the matrix results in a more heterogeneous system allowing for faster rotational and translational dynamics of the ionic species. All of these factors cumulatively result in a decreased $T_g$ of the solvent. This is consistent with the results of orientational and translational dynamics studied.

**Discussion**

Several different experimental and computational techniques were used to investigate how the microscopic interactions and local structure evolve in ChCl/glycerol mixtures as the HBA concentration is increased in the HBD. Neat glycerol has an

**Table 2 Average hydrogen bond dynamics time constants (ps) derived from CMD simulations at 300 K for 0, 5, 10, 20, and 33 mol% ChCl in Glycerol.**

| | Site-site | 0 mol% | | 5 mol% | | 10 mol% | | 20 mol% | | 33 mol% | |
|---|---|---|---|---|---|---|---|---|---|---|---|
| | | $\tau_f$ | $\tau_b$ | $\tau_f$ | $\tau_b$ | $\tau_f$ | $\tau_b$ | $\tau_f$ | $\tau_b$ | $\tau_f$ | $\tau_b$ |
| Gly–Gly | Oc–Hc | 182.1 | 497.7 | 160.2 | 435.4 | 148.1 | 396.5 | 122.9 | 330.2 | 98.1 | 261.7 |
| | Oc–Ht | 181.2 | 467.5 | 167.2 | 487.2 | 153.8 | 459.0 | 127.5 | 420.6 | 103.2 | 388.4 |
| | Ot–Hc | 191.0 | 483.0 | 172.5 | 442.6 | 160.0 | 436.7 | 137.6 | 397.3 | 119.2 | 395.7 |
| | Ot–Ht | 182.1 | 497.7 | 198.7 | 390.8 | 182.4 | 356.4 | 150.8 | 290.6 | 122.7 | 233.3 |
| Cl–Gly | Cl–Hc | – | – | 162.9 | 159.9 | 148.5 | 162.9 | 114.8 | 125.8 | 87.4 | 91.9 |
| | Cl–Ht | – | – | 234.7 | 130.5 | 209.7 | 116.9 | 170.6 | 90.0 | 131.6 | 66.0 |
| Cl–Ch | Cl–Hy | – | – | 179.6 | 178.9 | 162.7 | 160.7 | 118.2 | 111.2 | 89.5 | 79.4 |
| Gly–Ch | Oc–Hy | – | – | 98.7 | 201.6 | 87.1 | 202.3 | 67.6 | 150.6 | 52.2 | 115.2 |
| | Ot–Hy | – | – | 117.7 | 204.9 | 105.2 | 181.6 | 79.4 | 128.8 | 62.3 | 95.0 |
| Ch–Gly | Oy–Hc | – | – | 67.5 | 612.7 | 66.5 | 527.3 | 54.6 | 501.6 | 42.8 | 383.5 |
| | Oy–Ht | – | – | 74.4 | 758.0 | 69.6 | 784.1 | 56.1 | 673.4 | 45.6 | 601.9 |
| Ch–Ch | Oy–Hy | – | – | 6.4 | 36.2 | 13.7 | 70.3 | 15.1 | 93.0 | 12.6 | 85.1 |

$\tau_f$ is associated with hydrogen bond breaking and is equivalent to $\tau_{hb}$ which describes the average hydrogen bond lifetime. $\tau_b$ is the time constant of hydrogen bond reformation. Standard deviations of these values are provided in Supplementary Table 13.

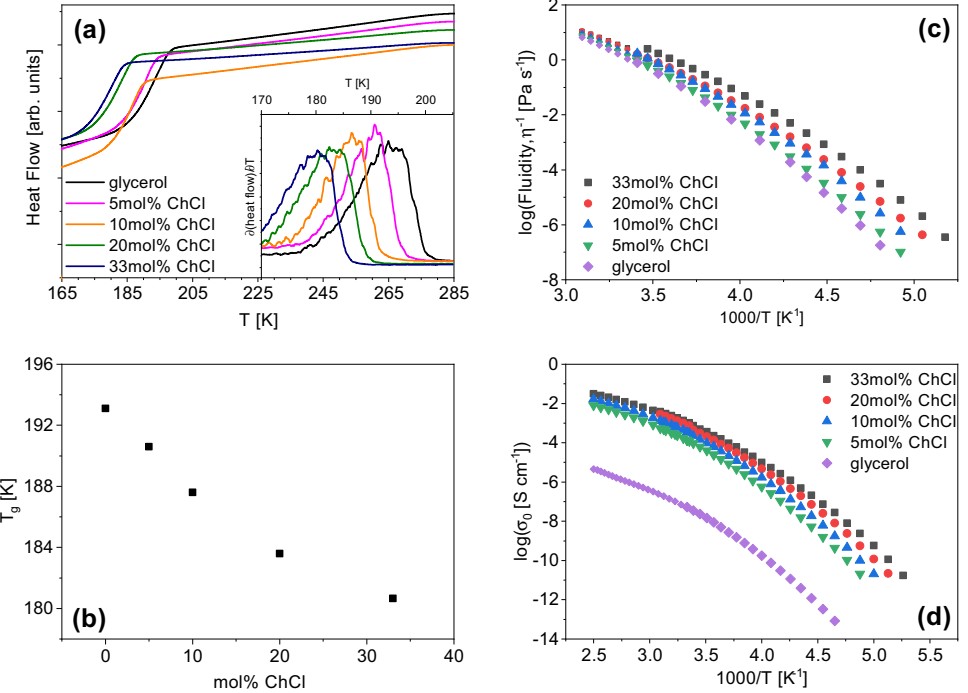

**Fig. 7 Physicochemical properties are enhanced in ChCl/glycerol mixtures as the eutectic composition is approached. a** Heat flow plotted versus temperature for 0–33 mol% ChCl for the cooling cycle obtained from DSC. Inset: The derivative of heat flow with respect to temperature plotted versus temperature. **b** Glass transition temperatures of the mixtures plotted versus the choline chloride concentration. **c** Fluidity, inverse viscosity, plotted versus inverse temperature for 0–33 mol% ChCl in glycerol. Measurements below 298 K were obtained with the rheometer described in methods under DMS, and above 298 K were made with a viscometer. **d** Dc ionic conductivities of 0–33 mol% ChCl in glycerol plotted versus inverse temperature.

extensive, branched HBN[40]. As ChCl is added, it is observed from ¹H NMR, that chloride donates electron density to glycerol as the two begin interacting. In addition, WANS supported by CMD and AIMD simulations show that these interactions alter and weaken the HBN of glycerol. This can be attributed to the finding that Gly–Cl interactions are similar, if not weaker, in strength than Gly–Gly interactions (Table 2). However, choline still strongly interacts with chloride at the eutectic composition but does not significantly interact with glycerol, shown by $N_{coord}$. Overall, this creates local, structural heterogeneity, which weakens the HBN of glycerol and drastically increases solvent dynamics.

The dynamic mechanical spectra in Fig. 4b suggest that glycerol mostly preserves much of its HBN even with 33 mol% ChCl present, since the shapes of the structural relaxation and the slow mechanical relaxation, originally attributed to the HBN of glycerol, remain unchanged. This indicates that though ions are dispersed throughout the system and decrease glycerol self-interactions, the majority of glycerol molecules remain interconnected in some form. CMD simulations of dipole moment rotational correlation relaxation times and diffusion coefficients, QENS, and PFG-NMR revealed that choline rotates and diffuses slower than glycerol at all concentrations and timescales

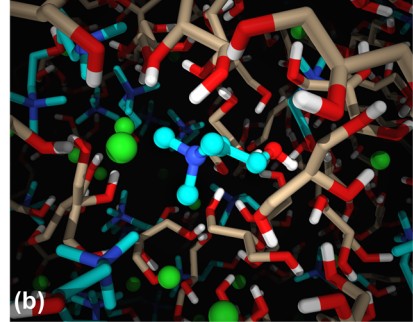

**Fig. 8 Snapshots of the environment of choline in 5 and 33 mol% ChCl in glycerol. a** Snapshot from a CMD simulation of slow reorienting choline in 5 mol % ChCl. The white–red–tan molecule represents glycerol and the hydrogen–oxygen–carbon atoms, respectively. The white–red–aqua-blue molecule represents choline and the hydrogen–oxygen–carbon–nitrogen atoms, respectively. The lime green molecule is the chloride anion. Choline is "trapped" within homogeneous glycerol HBN. **b** Snapshot from a CMD simulation of fast reorienting choline in 33 mol% ChCl. Heterogeneities of species and charge alternation weaken the HBN significantly, liberating choline cations to more easily reorient within the HBN.

examined. This is because the glycerol HBN acts as a cage to choline, which is weakened as ChCl concentration is increased. The simulated dipole moment rotational correlation relaxation times in Fig. 6a show that choline becomes faster relative to a fixed dynamic timescale of glycerol as ChCl concentration is increased, which is the same trend observed for the sub-α and α-dielectric relaxations, respectively, shown in Fig. 5b. This, along with the coupling of the sub-α relaxation to ion dynamics shown in Fig. 5c, indicates the sub-α dielectric relaxation is a reflection of the slow, rotational ion dynamics of choline in the DES, which are able to increase in rate as the system becomes more mobile.

The addition of ChCl to glycerol creates microscopic heterogeneities that are dispersed throughout the mixture. As ChCl concentration is increased up to the eutectic composition, the network becomes more heterogeneous, which weakens the HBN of glycerol thereby increasing the molecular mobility as observed from QENS, PFG-NMR, CMD simulations, and BDS. This presumably results in the characteristic depression in the glass transition temperature of Glyceline. In addition, since the transport and dynamic properties are coupled to the glass transition, the structural relaxation rates, fluidity, and conductivity all increase with ChCl concentration accordingly. These are direct results of the weakening, though not complete disruption, of the HBN of glycerol as ChCl is added. Thus, the evolution of microscopic heterogeneities and dynamics with increasing ChCl concentration is shown to correlate with the macroscopic transport properties in the ranges studied.

## Methods

**Sample preparation for protonated samples**. Glycerol and ChCl were purchased from Sigma Aldrich at ≥99% and ≥98% purity, respectively. EG was purchased from ACROS Organics at 99+% purity. Betaine-30 (Reichardt's dye, 90%, Sigma-Aldrich) was used as received for fs-TS measurements. Glycerol was used as received. The DES preparation and characterization was performed following protocols reported by Gurkan et al.[51] ChCl was dried in a vacuum oven for 2 days at 393 K. Appropriate amounts of ChCl were added to glycerol or EG, and stirred at 353 K in a dry nitrogen atmosphere until a clear liquid was formed. Samples prepared for [1]H NMR, BDS, DMS, fs-TA, and DSC were not prepared in a glove box, so they were briefly exposed to ambient conditions.

**Proton and pulsed-field gradient nuclear magnetic resonance spectroscopy**. [1]H NMR data were recorded on a Bruker Avance III spectrometer operating at 400 MHz using a direct 5-mm broadband probe. The sample temperature was established at 333 K with an equilibration period of 1 h before data acquisition. The chemical shifts were referenced to trimethylsilane as an external standard. [1]H NMR spectra were recorded with a standard one-pulse sequence with a 90° flip angle, with a spectral width of 20 ppm, and a relaxation delay of 1 s.

NMR diffusion experiments were performed with a 400 MHz Bruker spectrometer. PFG-NMR stimulated echo sequence was used to measure the self-diffusion coefficients of glycerol and choline independently (using their well-differentiated peaks in the NMR spectrum), at variable temperatures covering the 298–338 K range. The gradient strength, $g$, was varied linearly over 16 values ranging from 0 to 80 G cm$^{-1}$. The gradient pulse duration was $\delta = 7$–9 ms and the diffusion delay was $\Delta = 0.5$–1.2 s. The self-diffusion coefficients, $D$, were then extracted by fitting the decay echo signal with the Stejskal–Tanner equation: $I = I_0 \exp[-D(\gamma g \delta)^2(\Delta - \delta/3)]$, where $I$ is the amplitude of the attenuated echo signal, $I_0$ is the initial intensity and $\gamma$ is the proton gyromagnetic ratio[52].

**Classical molecular dynamics simulations**. Fully atomistic equilibrium molecular dynamics simulations were run using LAMMPS[53] with intra- and inter-molecular interactions described by the general AMBER force field[54]. Initial structures of individual glycerol and choline molecules were optimized in vacuum-phase using Gaussian 09[55] using a B3LYP/aug-cc-pvdz basis set. Atomic Lennard–Jones (LJ) sigma and epsilon values were adapted from a study by Perkins et al[33]. Atomic charges were also adapted from the same study, though the atomic charges of choline and chloride were uniformly re-scaled from the given 0.9–0.7 net molecular charge. Charges of glycerol atoms were unmodified. The studies indicated that charge scaling higher than 0.7 did not correctly model ChCl glycerol mixture dynamics, resulting in systems that became slower with added ChCl rather than faster. While a 0.7e scaling resulted in absolute bulk phase dynamics that were faster than experimental measurements, the correct physical trends were properly captured for the observed concentration and temperature range. The parameters used to describe ChCl and EG mixtures were detailed in our previous publication[34]. A 1 fs timestep, velocity Verlet integrator, periodic boundary conditions in all directions, cutoffs for LJ and Coulombic interactions of 12 Å, and a particle–particle particle-mesh long-range solver at $10^{-4}$ accuracy were used for all simulations.

Initial conformations were randomly and loosely packed into a cube via Packmol[56,57]. The composition of each simulation box is summarized in Table 3. Simulations for ChCl and glycerol mixtures were run at temperatures between 280 and 400 K in 20 K increments for a total of seven temperatures. ChCl and EG mixtures were simulated at 298 K. Boxes were equilibrated for 2 ns in the isothermal isobaric (NPT) ensemble at one atmosphere and their respective temperatures until the density and total energy had converged for at least 1 ns. Box dimensions were taken from the average of the last 1 ns of the converged NPT run for initializing a canonical (NVT) ensemble production run. Production runs were carried out for 20 ns. A Nosé–Hoover-chain thermostat and Nosé–Hoover-chain barostat with a chain length of 3 and time constant of 100 fs was used where applicable in all simulations. Further production runs specifically for viscosity calculations were initialized from the equilibrated NVT configurations following the method detailed by Zhang et al. with viscosity calculated via a time decomposition method using the integral over time of the pressure tensor autocorrelation function[58]:

$$\eta = \frac{V}{6 k_B T} \int_0^\infty \sum_{\alpha \le \beta} \langle \bar{P}_{\alpha\beta}(t) \cdot \bar{P}_{\alpha\beta}(0) \rangle \ \mathrm{d}t \qquad (1)$$

In the above equation, $V$ is the system volume, $k_B$ is the Boltzmann constant, $T$ is system temperature, and $\bar{P}_{\alpha\beta}$ is the modified pressure tensor of element $\alpha\beta$. To improve statistics, viscosities are averaged over the six correlation functions of the modified pressure tensors: $\bar{P}_{xy} = P_{xy}$, $\bar{P}_{yz} = P_{yz}$, $\bar{P}_{xz} = P_{zx}$, $\bar{P}_{xx} = 0.5(P_{xx} - P_{yy})$, $\bar{P}_{yy} = 0.5(P_{yy} - P_{zz})$, $\bar{P}_{zz} = 0.5(P_{xx} - P_{zz})$. $P_{\alpha\beta}$ are elements of the standard pressure tensor. Viscosity calculations were carried out using the PyLAT tool[59].

**Table 3 Composition of simulation boxes.**

| ChCl (mol%) | Number of ChCl | Number of HBD | Number of total atoms |
|---|---|---|---|
| *Glycerol* | | | |
| 0 | 0 | 715 | 10,010 |
| 5 | 35 | 665 | 10,080 |
| 10 | 68 | 612 | 10,064 |
| 20 | 129 | 516 | 10,062 |
| 33 | 200 | 400 | 10,000 |
| *Ethylene glycol* | | | |
| 0 | 0 | 1000 | 10,000 |
| 5 | 100 | 1900 | 21,200 |
| 10 | 150 | 1350 | 16,800 |
| 16.7 | 150 | 750 | 10,800 |
| 20 | 175 | 700 | 10,850 |
| 25 | 200 | 600 | 10,400 |
| 33 | 250 | 500 | 10,500 |

Site–site RDFs, coordination numbers ($N_{coord}$), combined distribution functions (CDFs), and hydrogen bond dynamics were calculated using the TRAVIS program[60]. Self-diffusion coefficients were calculated using the Einstein expression in Eq. (2) and the MSD was calculated using Eq. (3).

$$D_S = \frac{1}{6N} \lim_{t \to \infty} \frac{d}{dt} \sum_{i=1}^{N} \langle [\mathbf{r}_i(t) - \mathbf{r}_i(0)]^2 \rangle \qquad (2)$$

$$\langle [\mathbf{r}_i(t) - \mathbf{r}_i(0)]^2 \rangle = \frac{1}{n+1} \sum_{t_0=0}^{n} [\mathbf{r}_i(t_0 + t) - \mathbf{r}_i(t_0)]^2 \qquad (3)$$

where $N$ is the number of individual species and $\mathbf{r}_i$ is the center of mass position of the $i$th species. The dipole moment was defined for each molecule by calculating the positive and negative centers of charge, using only positive and negative atoms respectively, and drawing a vector between the two for each molecule. A dipole autocorrelation was calculated using the correlation function

$$C(t) = \left\langle \frac{1}{2N_i} \sum_{i=1}^{N_i} [3 \cos^2 \theta_i(t) - 1] \right\rangle \qquad (4)$$

where $N_i$ is the number of such vectors, and $\theta_i(t)$ is the angle between each vector at time $t = 0$ and the same vector at time $t$. The decay behavior of the correlation functions with ChCl present exhibited both fast and slow modes not easily fit a pure exponential function, possibly due to non-integer dynamics at fast time scales. Examples of the correlation functions are shown in Supplementary Fig. 16. To model this, the following functional form was used

$$C(t) = b_1 E_{\alpha 1}(-(t/\tau_1)^{\alpha_1}) + b_2 E_{\alpha 2}(-(t/\tau_2)^{\alpha_2}) \qquad (5)$$

where $z = t/\tau_n$ and $E_\alpha(z)$ is the Mittag–Leffler function

$$E_\alpha(z) = \sum_{n=0}^{\infty} \frac{z^n}{\Gamma(\alpha + 1)} \qquad (6)$$

also expressed as the Laplace transform[61,62]

$$E_\alpha(-z^\alpha) = \frac{1}{\pi} \int_0^\infty e^{-xz} \frac{x^{\alpha-1} \sin \pi\alpha}{x^{2\alpha} + 2x^\alpha \cos \pi\alpha + 1} dx, 0 < \alpha < 1 \qquad (7)$$

where $b_n(b_1 + b_2 = 1)$, $\alpha_n$, and $\tau_n$, $n = 1, 2$, are fitting parameters, $\tau_n$ describes relevant time scales of relaxation, and $\Gamma(x)$ represents the Gamma function. Note that a Mittag–Leffler function reduces to an exponential function when $\alpha = 1$. When $\alpha$ are non-integer values, $E_\alpha(z)$ functions as a stretched exponential for small values of $z$, $E_\alpha(-(t/\tau)^\alpha) \approx \exp(-(t/\tau)^\alpha)/\Gamma(1+\alpha)$, which then asymptotically decays following a power law $(t/\tau)^{-\alpha}$.

Simulated neutron scattering structure factors, $S(q)$, were computed by

$$S(q) = \frac{\rho_0 \sum_i \sum_j x_i x_j f_i f_j \int_0^{\frac{L}{2}} 4\pi r^2 (g_{ij}(r) - 1) \frac{\sin(qr)}{qr} W(r) dr}{[\sum_i x_i f_i]^2} \qquad (8)$$

where $\rho_0$ is the total count density, $x_i$ and $x_j$ are the mole fractions of atoms $i$ and $j$, $f_i$ and $f_j$ are the appropriate atomic neutron scattering form factors[63] of atoms $i$ and $j$, $L$ is the simulation box length, $g_{ij}(r)$ is the RDF between atoms $i$ and $j$, and $W(r) = \sin(2\pi r/L)/(2\pi r/L)$ is a Lorch window function accounting for finite-size truncation of the $g_{ij}(r)$ at large distances.

**Ab initio molecular dynamics simulations.** Initial atomic coordinates for the Glyceline AIMD simulation (16 ChCl ion pairs with 32 glycerol molecules) were acquired from a 10 ns CMD simulation in the canonical NVT ensemble using the simulation package PINY_MD[64] and a force field that was previously developed for 8 unique DES systems[65]. The starting geometry for the CMD simulation was constructed

in a cubic box with the program Packmol[57]. A box edge length of 19.7 Å was kept constant to reproduce extrapolated experimental densities at 400 K[66]. In order to alleviate the high calculation costs of an AIMD simulation, multiple time stepping techniques (MTS) have previously been developed[67,68]. Of these MTS methods, the reversible reference system propagator algorithm (r-RESPA)[69] was utilized. In this approach, the force of the system $F(\mathbf{R})$ is separated into an inexpensive reference-system force $F_{ref}(\mathbf{R})$, that serves as a crude approximation of $F(\mathbf{R})$, and a correction $F_{corr}(\mathbf{R})$ such that $F(\mathbf{R}) = F_{ref}(\mathbf{R}) + F_{corr}(\mathbf{R})$. Both $F_{ref}(\mathbf{R})$ and $F_{corr}(\mathbf{R})$ are treated with different time steps and levels of theory while retaining time reversibility and numerical stability. Recently, this approach has been successfully applied to liquid imidazole using a combination of density functional tight-binding (DFTB)[70,71] as a reference system and corrections derived from full density functional theory (DFT)[72]. The success of this recent study shows that r-RESPA can be integrated into the AIMD framework and can be applied to DESs as well[73].

AIMD simulations using r-RESPA were carried out using DFT and the 3rd order variant of the density functional tight-binding method (DFTB3) within the CP2K package utilizing the QUICKSTEP module[74]. For the DFT correction, forces and energies of all atoms of the system were calculated using a hybrid Gaussian and plane wave approach with the molecular optimized triple-basis set (MOLOPT-TZVP-GTH)[75], along with the Perdew–Burke–Ernzerhof functional and the corresponding Goedecker–Teter–Hutter psuedopotentials[76] with an empirical dispersion correction (D3)[77]. A density cutoff of 520 Ry was used. The finest grid level was set with a multigrid number 4 and a relative cutoff of 60 using the smoothing for the electron density (NN10_SMOOTH) and its derivative (NN10)[74]. To calculate $F_{ref}(\mathbf{R})$, DFTB3 was used in conjunction with the 3ob-3-1 parameters[78,79] and the D3 dispersion correction. The trajectory was sampled using periodic boundary conditions and the box was equilibrated for 10 ps with the canonical NVT ensemble at 400 K held using the Nosé–Hoover chain thermostat[80,81]. Production runs were carried out for an additional 25 ps. The DFTB3 reference system was integrated with a time step of 0.5 fs and the DFT corrections were applied every 2.0 fs. RDFs and CDFs were calculated using the program TRAVIS[60].

**Wide-angle and quasi-elastic neutron scattering.** WANS experiments were performed at Oak Ridge National Laboratory on the nanoscale ordered materials diffractometer over a scattering vector range of 0.1 to 30 Å$^{-1}$. Mixtures of deuterated ChCl (trimethyl-d9, 98%) and deuterated glycerol (d8, 99%) at ratios of 1:2 (33% ChCl) and 1:20 (5% ChCl) were formed in a glove box with ≤0.2 ppm $O_2$ in 3 mm quartz capillaries. Both deuterated ChCl and deuterated glycerol were obtained from Chemical Isotopes Laboratory. Multiple neutron diffraction data sets of each composition were measured in 30 min intervals and summed to improve statistics. Reduction of data, quartz background scattering, and vanadium normalization was then performed using Addie[82] to obtain the structure factors, $S(q)$. All neutron diffraction measurements were performed at 298 K.

For the QENS experiment, four ChCl and glycerol mixtures were prepared in a controlled atmosphere glove box. Two samples were prepared in the ratio of 1:9 (10% ChCl, 90% d-glycerol, and 10% d-ChCl, 90% glycerol) and the other two samples were mixed in the ratio of 1:2 (33% ChCl, 67% d-glycerol, and 33% d-ChCl, 67% glycerol). Prior to mixing, ChCl was heated for 2 h at 393 K. After the addition of the appropriate amount of glycerol with ChCl, the mixtures were kept at 353 K for an additional 2 h for equilibration. Elastic fixed window scans were completed on 10% and 33% ChCl samples using the High Flux Backscattering Spectrometer (HFBS) at the National Institute of Standards and Technology (NIST) Center for Neutron Research (NCNR)[83]. The HFBS has an instrument resolution of ~0.8 μeV (about 2 ns) and elastic intensity was measured over the $q$ range of 0.25–1.75 Å$^{-1}$ in the temperature window from 4 to 298 K with a temperature ramp rate of 0.8 K/min. The elastic intensity was measured on heating and cooling to verify the reversibility of the dynamics and that the components did not crystallize during the process. The measurements were reduced and analyzed using DAVE, a data analysis and visualization software[84].

**Broadband dielectric spectroscopy.** BDS investigates the polarization response to an alternating electric field over a wide frequency and temperature range. This response may be given in terms of the real and imaginary parts of complex permittivity, $\varepsilon^*(\omega) = \varepsilon'(\omega) - i\varepsilon''(\omega)$, which can be directly related to complex conductivity as $\varepsilon^*(\omega) = i\omega\varepsilon_0\sigma^*(\omega)$, where $\omega$ is the radial frequency and $\varepsilon_0$ is the permittivity of free space. The complex conductivity function can also be expressed in terms of the real and imaginary parts as $\sigma^*(\omega) = \sigma'(\omega) + i\sigma''(\omega)$. Since $\varepsilon''(\omega)$ is directly related to the real part of conductivity, $\sigma'(\omega)$, dielectric relaxations can become obscured in ion-conducting materials due to the dominant contributions of the dc ionic conductivity. Therefore, we instead examine the derivative representation, $\varepsilon''_{der} = -\pi/2(\partial\varepsilon'/\partial\ln(\omega))$, to unveil any relaxations which may be overshadowed by contributions from the ionic conductivity.

A Novocontrol High-Resolution Alpha Dielectric Analyzer in combination with a Quatro cryogenic temperature control system was used to perform the dielectric measurements from $10^{-1}$ to $10^7$ Hz from 150 to 400 K, with accuracy better than ±0.1 K. The samples were measured in parallel plate capacitor geometry with 20 mm diameter gold-plated brass electrodes. In addition, an Agilent E5071C Vector Network Analyzer was used for room temperature measurements from $3 \times 10^5$ to $2 \times 10^{11}$ Hz. The dielectric data for glycerol in the high-frequency regime were

adequately fit using a single Havriliak–Negami function. However, the data for the ChCl/glycerol and ChCl/EG mixtures were fit using a combination of a Debye function, Havriliak–Negami function, and a Random Barrier Model. The three functions for the mixtures were necessary to account for contributions from the slow, structural $\alpha$, and ion dynamics, respectively. These fits are outlined in the SI, as Supplementary Eqs. 3 and 4.

**Dynamic mechanical spectroscopy.** Oscillatory-shear rheology measurements of 0, 5, 10, 20, and 33 mol% ChCl in glycerol were made using a TA Instruments Discovery Hybrid Rheometer-2. Parallel plate geometry was used with 8 mm for higher temperatures and 3 mm plates for measurements at temperatures approaching the glass transition. The dynamic mechanical spectra were obtained from $10^{-1}$ to $10^2$ Hz at temperatures ranging from 178 to 223 K. Time–temperature superposition was employed to obtain a master curve for each composition investigated. The data were fit using two Cole–Davidson modified Maxwell equations, provided in Supplementary Eq. 5[85]. Viscosity measurements below 298 K were made on the TA Instruments Discovery Hybrid Rheometer-2 by applying a constant shear rate.

**Femtosecond transient absorption spectroscopy.** The samples of glycerol and mixtures with ChCl were cooled from 353 K after mixing to 303 K and transferred into sealed cuvettes to prevent moisture ingress. Femtosecond transient absorption spectroscopy was performed on a ClarkMXR CPA-2001 laser, outputting a 780 nm fundamental wavelength, at a repetition rate of 1 kHz. This output beam was split, with 5% going to a sapphire crystal to create the supercontinuum probe beam, and the rest to a light conversion TOPAS to produce a 490 nm wavelength pump beam. Pump power was chosen as the lowest power that provided a stable signal, which was 40 mJ cm$^{-2}$. Steady-state UV-vis measurements were performed using a Varian Cary 50 UV–vis Spectrophotometer. Data were chirp-corrected and fit to biexponential decay functions, modeled by: $\Delta Abs = A_1 \exp{(\tau_1 t)} + A_2 \exp{(\tau_2 t)}$, where $A_1$ and $A_2$ are amplitude coefficients, $t$ is time, and $\tau_1$ and $\tau_2$ are time constants.

It is worth pointing out that preferential solvation to betaine-30 probes has been observed in past experiments[86–88]. Indeed, apolar probes elicit hydrophobic solute-solvent interactions (Van der Waals type interactions), while polar groups lead to the possible induction of electronic polarization, ionic interactions, and also possible hydrogen bonding. The solvent systems studied here are of highly polar character, with dominant hydrogen-bonding characteristics. Thus, an apolar probe molecule might have led to different results and different relaxation components. This is why we have chosen betaine-30 as the probe molecule, which has a pronounced charge-transfer character in its ground state. The charge transfer characteristic implies that the molecule is strongly surrounded by the HBD molecules of the DES, as well as the ionic species, as suggested above. Furthermore, the molecular simulations in this work suggest a strong hydrogen bonding network in the bulk solvent. The similarity in solvent-solvent interactions in the bulk and the solvent-solute interactions in the surrounding of the probe molecule explains the observed quantitative agreement in the mean rates of the dynamics obtained from fs-transient absorption measurements, BDS, and dynamic molecular simulation data.

**Differential scanning calorimetry.** A TA Instruments Q2000 Differential Scanning Calorimeter was used to determine the calorimetric glass transition temperatures. The measurements were obtained by cycling from 298 to 163 K and back up to 298 K twice at 10 K/min. No melting or crystallization was observed.

## Data availability

The authors declare that the data supporting the findings of this study are available within the article and its Supplementary Information. Additional data is available from the corresponding authors upon request.

## Code availability

The computer code used for simulations is available from the corresponding authors upon request.

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

## Acknowledgements

This work was supported as part of the Breakthrough Electrolytes for Energy Storage (BEES), an Energy Frontier Research Center funded by the U.S. Department of Energy, Office of Science, Basic Energy Sciences under Award #: DE-SC0019409. D.P., Y.Z., and E.M. thank the Center for Research Computing (CRC) at the University of Notre Dame for providing computational resources. S.S., T.C., and J.S. thank Yangyang Wang at Oak Ridge National Laboratory for instrument use in the Center for Nanophase Materials Sciences. Access to HFBS was provided by the Center for High Resolution Neutron Scattering, a partnership between the NIST and the National Science Foundation under agreement no. DMR-2010792. The identification of any commercial product or trade name does not imply endorsement or recommendation by NIST.

## Author contributions

S.S., T.C., and J.S. performed the broadband dielectric spectroscopy, differential scanning calorimetry, and dynamic-mechanical spectroscopy experiments and analyses. D.P., Y.Z., and E.M. performed the classical molecular dynamics simulations and analyses. B.D. and M.T. conducted the ab initio molecular dynamics simulations and analysis. R.E. and T.Z. performed the $^1$H NMR experiments and analyses. C.F. and S.G. performed the field-gradient NMR experiments and analysis. H.S. and B.G. conducted the density, viscosity, and ionic conductivity measurements and analyses. L.H., M.A.H., M.T., and M.D. performed the neutron scattering experiments and analyses. C.K. and C.B. provided the femtosecond transient-absorption spectroscopy experiments and analysis. S.S., D.P., B.D., E.M., and J.S. wrote the paper with input from all authors. J.S., E.M., M.T., T.Z., and M.D. conceived of the project and were responsible for the overall project management.

## Competing interests

The authors declare no competing interests.
