## [Peer Review File · Nature Communications]

Reviewers' comments:

Reviewer #1 (Remarks to the Author):

I have read this paper dealing with a multi-technique approach to investigate dynamics and micro-structure of DESs. I think the approach and results are sound but the main claim that "These results provide a unified framework for rationalizing the key features of DESs" is a bit of an overstatement. The work is focused on a single DES only (Glyceline) and only looking at two ChCl compositions, and to validate such a claim a more extensive study on at the least several types of DESs should be conducted. In addition, it is not clear what the real breakthrough of this work is. Correlating dynamics with macroscopic behaviour sounds a bit vague. My comments are below:

- A single DES (Glyceline) is chose for this study, it is therefore a bit risky to draw conclusions that applies to the whole category of these liquids.

- In addition to my previous comments, only two concentrations are studies, 5% and 33% of ChCl. The mixture is an eutectic so it is expected that by a first increase in concentration of ChCl the glass transition temperature decreases, fluidity increases as well as ionic conductivity (all expected behaviours). Likewise I would expect a similar behaviour by approaching the eutectic point starting from 100% ChCl, however, I see that the range of concentration studied does not cover this part of the eutectic and I wonder why this range is not being considered.

- Figure 1, the spectrum reports a resonance labelled as "d", this is however not reported on any of the molecular structures nor explained in the caption. This needs to be revised.

- I cannot really see the slight shoulder of peak "i" in Figure 1, bottom.

- Figure 1, the 5mol% ChCl has much boarder peaks than the 33mol% ChCl. In general, from what I understand from the author's discussion higher viscosity means broader peaks, does that mean that the 5mol% is more viscous than the 33mol%, hence showing broader peaks? I assume this is the case but it should be clarified.

- It is not clear what Ncoord is, I understand is a coordination number but of what species or group within the molecule? Also, the values are below 1 in general, and with decimal, so I assume this does not refer to number of actual atoms or molecules surrounding the species under investigation?

- In the abstract it says " We find that the microscopic heterogeneities induced in glycerol by the addition of the choline chloride lead to new slow, dynamic modes that are strongly correlated to the macroscopic properties of the DES formed". However, I understand from the text and from Figure 8 that the dynamic becomes faster? This is a bit confusing.

- I find the paper difficult to read in some points. I understand that a lot of techniques and results are included but the way these are presented makes it not easy to follow. The paper would benefit by breaking it down into different sub-sections discuss the results of each technique or group of techniques and conclude highlighting the main finding for each technique.

Reviewer #2 (Remarks to the Author):

Title: Correlating Microscopic Heterogeneity and Dynamics in Deep Eutectic Solvents

Authors: Stephanie Spittle, Derrick Poe, Brian Doherty, Charles Kolodziej, Luke Heroux, Md Ashraful Haque, Henry Squire, Tyler Cosby, Yong Zhang, Carla Fraenza, Sahana Bhattacharyya,

Madhu Tyagi, Jing Peng, Ramez A. Elgammal, Thomas Zawodzinski, Mark Tuckerman, Steve Greenbaum, Burcu Gurkan, Clemens Burda, Mark Dadmun, Edward J. Maginn, and Joshua Sangoro.

The authors present a detailed study on Glyceline (as a model Deep Eutectic Solvent – DES) in order to correlate local structure to macroscopic dynamics. A wide range of computational and experimental techniques covering length-scales from molecular to macroscopic and time-scales from picoseconds to seconds are used for this purpose. They found that the microscopic heterogeneities induced in glycerol by the addition of the choline chloride lead to new slow, dynamic modes that are strongly correlated to the macroscopic properties of the DES, essentially, the decreased glass transition temperature and a corresponding increase in dc ionic conductivity, fluidity, diffusivity, and orientational dynamics (computed and/or measured in this work).

Novelty and key results:

The work mainly focuses on investigating the evolution of the heterogeneities in the selected DES with composition and their influence on the properties of the DES. The work is novel and relevant to the field of DES in particular and alternative solvents in general. The key result is the finding of the new slow, dynamic modes when ChCl is added to glycerol, their relationship with the microscopic heterogeneities and with the macroscopic properties of the DES, supported by different experimental and modeling techniques. The key scientific questions posed by the authors in the introduction are relevant and properly addressed in the manuscript. However, the third question “are there dynamic modes that are unique to DESs and if so, what microscopic mechanisms dominate such processes?” is not sufficiently answered, as the study is just focused in one DES. It is implicitly assumed in the manuscript that the findings will hold for other DESs (see title and discussion), although results for other DES are not presented.

The manuscript is well written and clear. It provides strong evidence for its conclusions. The methodology is appropriate and impressive, considering the different techniques used and the time and length scale they cover. The quality of the data and quality of presentation are good (some specific details for improvement are provided next). The topic fits well into scope and audience of Nature Communication. However, the work needs major revisions before it can be accepted. Here are my comments and suggestions I would like the authors to address in their revised version:

General comments:

1. Title and abstract: Although a wide range of complementary techniques have been used in this work, it is just focused in one model DES (Glyceline). The title is too generic, it should be revisited to accordingly reflect the work presented in the manuscript, specifying the effect of concentration and the DES under study. Also, the sentence “These results provide a unified framework for rationalizing the key features of DESs including a decreased glass transition temperature and a corresponding increase in dc ionic conductivity, fluidity, diffusivity, and the mean rates of orientational dynamics” sounds too ambitious. The procedures used may allow doing it, but not the results, as they only apply to glyceline.

2. The authors have studied the effect of adding ChCl at different concentrations, up to the eutectic concentration in glycerol by different techniques; different temperatures were also chosen, depending on the technique to be used. The selected values of the ChCl concentration are quite consistent among the different methods used (although in some cases only one or two concentrations have been considered); however, different techniques have been used at different temperatures (ranging from 215K to 400K) in a less consistent manner than the concentrations. In fact, it is difficult to follow the effect of temperature with the different techniques used, although, for sure, temperature clearly affects both, the dynamics and the heterogeneity. This is an important issue to be clarified by the authors, and some rationale behind the selections and implications should

be presented and discussed. Notice that although some experimental techniques will work better at specific temperatures, simulations can be performed at the same temperatures as the experiments.

More specific comments (by order of appearance in the text)

3. Lines 33-35: it is claimed that little is known about the fundamental relationship between their molecular structure and the macroscopic properties of DES. I agree that considerable work needs to be done to further advance in the field; however, this statement is too strong and not completely accurate, as several experimental and modeling works on DES have been published already, providing very valuable insights into this relationship, as highlighted later in the text with some specific properties/methods. The sentence should be revisited.

4. Line 50: it should be "HBA" instead of "HBD" when referring to ChCl.

5. Line 74: please, define QENS

6. Line 95: The ^1H NMR spectra were obtained at 5 and 33 mol% concentration of ChCl in glycerol (at 333K), while simulations were also performed at 0 mol%, i.e., pure glycerol. Why there are not experimental results from pure glycerol presented? CMD results show the largest deviations for the density versus concentration at pure glycerol (Figure S1).

Also, related to Figure 1, the authors forgot to mention what the peak "d" represents.

7. Regarding Classical Molecular Dynamics Simulations (CMD) and their comparison with experimental data:

- The first comparison is shown in figures S1 and S2 in the supplementary information

Figure S1: Density as a function of mol% Choline Chloride in Glycerol at 300K for CMD and experimental methods. Although deviations are not large, the slope of the experimental density vs. mol% is different than the one from CMD, being the difference larger near the eutectic concentration. How relevant will the deviations in the density be for the conclusions made? Have the authors check any other force field?

Figure S2: Viscosity as a function of mol% Choline Chloride in Glycerol at 300K for CMD and experimental methods. The viscosity calculated with CMD follows the same trend as the experimental data; however, the obtained values are very different, being the simulated ones three

times smaller. It is also notice that the problem may arise from the viscosity calculated for glycerol (0% ChCl concentration), where the differences are larger.

- One more point related to the CMD: as the comparison with AIMD is only at 400K and 33 mol% ChCl in glycerol, how accurate is the FF used in CMD for densities and viscosities at 400K? (Figures S1 and S2 show only results at 300K)

8. Page 7: Figure 2: Regarding the RDF and CDF: results are presented just for 33 mol% ChCl in glycerol at 400K (Figure 2). Results are very interesting and clear with this representation; can you comment on the differences observed at other ChCl concentrations and lower temperatures (they should be available at least for the CMD simulations*)? I suggest showing a comparison and the corresponding discussion in the revised version of the manuscript, as both, composition and temperature are relevant to the focus of this work.

Also related to Figure 2: Units for the RDF are in Å (2b) while for the CDF (Combined Distribution Functions) are in pm (picometer) (2c). The units between both figures should be consistent, for an easier visual comparison of the results.

*Note: AIMD were done only at one temperature and one ChCl concentration

9. Lines 144-149: The authors state: "AIMD simulations at 33mol% indicate that the chloride anion does not selectively coordinate with the Hc or Ht hydrogens of glycerol, with Ncoord of 0.62 (Hc) and 1.27 (Ht), respectively, considering the 1:2 ratio of Hc:Ht in glycerol. This behavior is also seen in CMD simulations at 400 K. These results agree with the 1H NMR data at 333 K in Figure 1, which show a uniform shift of the Hc (e) and Ht (a) peaks as ChCl concentration is increased."

The 1H NMR data was obtained at 333K, while the CMD simulations were performed at 300 K and 400K (and AIMD only at 400K). The authors comment that the simulations at 400K agree with the 1H NMR data at 333K; however, different results are obtained at 300K. As 300K is closer to 333K than 400K, simulation results at this temperature are expected to better represent the experimental results. Can you further elaborate on this, please? Have you performed any simulations at 333K? it would be very interesting to see the comparison at the same temperature, and also the effect of temperature with more than two values.

10. Page 9, Table 1 includes the Coordination numbers (Ncoord) for ChCl in glycerol from CMD (300 and 400K, 5 and 33 mol% ChCl) and AIMD (400K, 33mol% ChCl).

- What are the error bars or uncertainties associated to the numbers provided in the table?

- The authors claim that (lines 155-156) at 300K, as the ChCl "concentration is increased, a reduced coordination of the oxygen atom on choline by glycerol is observed through a decrease in Ncoord of Oy-Hc and Oy-Ht"; however, a close look at the values provided in the table show similar numbers at the two temperatures and concentrations (depending on the uncertainties associated to these numbers, which are missing). On the contrary, a clear decrease in coordination numbers of Cl-Hc and Cl-Ht is observed as the concentration increases at 300K. Can you comment on this?

11. Line 168: The calculated results from CMD simulations at 0, 5, and -- mol% ChCl are also provided. The sentence is missing number "33" before "mol%"

12. Page 10, Figure 3: the structure factor at 0 and 5 mol% ChCl from CMD are very similar, and, as the authors pointed out, CMD agrees well with WANS, although the peak at 5 mol% is much more pronounced in the experiments than in CMD. In addition, a "shoulder" is observed at low q in the simulations at 33 mol% that does not appear in the experiments; also, there are two peaks between 2-4q that only appear in the simulations. By looking at Figure 3b it seems to be related to the strong Ch-Ch interactions. Can you, please, comment on that?

13. Figure 4

- The derivative representation of the dielectric loss is presented in Figure 4a for pure glycerol and 5 mol% ChCl, at different temperatures, ranging from 215-245K. All these temperature values are much lower than any of the values considered in the results previously presented in the manuscript. This goes back to my original comment on the different values of temperatures used for different techniques.

- Figure 4b: temperature?

14. Figure 5: the fact that the mean rates of orientational dynamics obtained from the fs-TA, BDS, and DMS are all in quantitative agreement is a remarkable result. Please, indicate what is the meaning of the dashed black line in Figure 5b.

15. Line 237: It is stated that: "Figure 5c shows that the sub-relaxation rate (ω_{slow}) is coupled to the ion hopping rate (ω_{ion}), and both are decoupled from ω_{α} . This suggests that the sub-dielectric relaxation has a similar origin to the ion dynamics, and both rates increase in identical increments with increasing ChCl concentration". This conclusion contradicts those of Faraone et al. and Reuter et al. However, a close look at Figure 5c (within the scale presented there) shows that after a certain concentration, ω_{α} seems to be coupled to the ion hopping rate (ω_{ion}), which goes against the argument of the two rates to be decoupled. Is it a problem of the scale shown in the figure? Please, comment on this.

16. Page 15 –Why there are just 2 points for choline E-S in Figure 6c?

17. Lines 300-301: it is stated that: "There is a reduction in both τ_f and τ_b in all pair-wise interactions with increasing ChCl save for Oy-Hy interactions that are not altered from the already low values". This statement is not completely correct, a look at Table 2 shows that, as expected, both τ_f and τ_b increase for Oy-Hy as the ChCl concentration increases, although their values are smaller than any

other O-H values. Interestingly, the average values are smaller at the eutectic concentration than at 20 mol%.

18. Figure 7 b. Caption: "(b) Fluidity, inverse viscosity, plotted versus inverse temperature for 0-33mol% ChCl in glycerol. Measurements above 298 K were done by another group." What do you mean? Are the results from the other group already published (if so, please, add the reference), if not, some more information should be provided (at least in the acknowledgements).

19. Line 366: "The addition of ChCl to glycerol creates heterogeneities that are dispersed throughout the mixture". These are microscopic heterogeneities, please, specify.

20. Lines 368-379: "As ChCl concentration is increased up to the eutectic composition, the network becomes more heterogeneous, which weakens the HBN of glycerol, and increases the molecular mobility of all components, observed from QENS, PFG-NMR, CMD simulations, and BDS. This results in the characteristic depressed glass transition temperature of Glyceline." Although the results point in this direction, this, by itself, is not the justification of the presence of the eutectic point, but a consequence of weakening the HBN, also observed in aqueous mixtures. Can you, please, further elaborate on this, and also comment on the influence of temperature?

21. Methodology: 421-422 Simulations were run at temperatures between 280 K and 400 K in 20 K increments for a total of seven temperatures. However, only results at 300 and 400K are presented and discussed. It will be of relevance to present at least the results at 333K, as already mentioned.

22. Lines 525-528: references are missing.

Lourdes F. Vega

Reviewer #3 (Remarks to the Author):

Deep eutectics are interesting materials, and this work focuses on glycerol/choline chloride at various compositions using a wide array of techniques, including experimental and computational approaches. Key observations are the reduced glass transition and the concomitant increase in numerous transport coefficients.

While an increase in heterogeneity by concentration fluctuations is a natural consequence of mixtures, two distinct peaks are observed via the analysis of dielectric BDS data and discussed as indication for pronounced heterogeneity as the eutectic composition is approached.

The amount of data collected for this system is commendable, but I am not convinced that all conclusions are supported by the results. For instance, the possibility of specific solvation (solvation largely by one component within a mixture) of Reichardt's dye is not considered. My main concern is in regards to the sub-alpha dielectric process, which seems to be visible already in pure glycerol,

according to figure 5b, lowest curve. According to the supplementary material figures, that peak position coincides with the condition where real and imaginary part of permittivity are equal in magnitude, which is known to indicate artifacts due to partly blocked electrodes. This sub-alpha peak is the main pillar of the strong heterogeneity claimed for this mixture, but it may not be a feature intrinsic in the eutectic. Also, the data of Faraone has less indication of such a peak, even if analyzed in the same manner. This component of the results requires careful re-consideration prior to publication.

Response to the Reviews and Decision

Title: Evolution of Microscopic Heterogeneity and Dynamics in Choline Chloride-based Deep Eutectic Solvents

**Manuscript Reference Number:
NCOMMS-20-24384**

Authors:

Stephanie Spittle
Derrick Poe
Brian Doherty
Charles Kolodziej
Luke Heroux
Md AshrafuHaque
Henry Squire
Tyler Cosby
Yong Zhang
Carla Fraenza
Sahana Bhattacharyya
Madhu Tyagi
Jing Peng
Ramez A. Elgammal
Thomas Zawodzinski
Mark Tuckerman
Steve Greenbaum
Burcu Gurkan
Clemens Burda
Mark Dadmun
Edward J. Maginn,
Joshua Sangoro

Date: April 6, 2021

Response To Reviewer #1

Overall Comments

I think the approach and results are sound but the main claim that "These results provide a unified framework for rationalizing the key features of DESs" is a bit of an overstatement. The work is focused on a single DES only (Glyceline) and only looking at two ChCl compositions, and to validate such a claim a more extensive study on at the least several types of DESs should be conducted. In addition, it is not clear what the real breakthrough of this work is. Correlating dynamics with macroscopic behaviour sounds a bit vague.

Response

We agree with the reviewer that some of our previous claim of generality was too broad given that we only studied glycerol/ChCl based systems. We have expanded our studies to ethylene glycol based systems as well and we demonstrate that our original conclusions are valid for this series of DESs as well. In addition, we have revised the title and abstract (line 13) to show that these conclusions apply for the DESs studied but could apply to other systems as well.

Action:

- Page 1: Edited title to properly reflect work done and materials studied
- Page 2: Lines 10-11 modified to include information about Ethaline series studied

Reviewer Comment

A single DES (Glyceline) is chosen for this study, it is therefore a bit risky to draw conclusions that apply to the whole category of these liquids.

Response

We agree with the reviewer that conclusions in our previous manuscript were based on studies of a single series of DESs and therefore could be misleading. Therefore, we have provided extensive results for another canonical DES, Ethaline, as a comparison. We show that the general conclusions drawn for the evolution of structure and dynamics in glycerol series apply to the ethylene glycol series as well. Evidence of the slow dynamic modes are demonstrated for this series both from our experimental and computational studies. Furthermore, the slow modes are found to arise from the ionic groups, implying that the modes arise from ion reorientations, suggesting that this is a general feature of Type III DESs.

In Figure 5, the derivative representation of the dielectric loss is shown as a function of frequency for various concentrations of ChCl in EG from 5-33mol% at single temperature. The same trend observed in Figure 4b for ChCl/glycerol mixtures is also observed in ChCl/EG mixtures, where the sub- α relaxation becomes faster relative to the structural relaxation as ChCl concentration is increased (discussed in lines 257-258).

In Figure S13, the ^1H NMR spectra of neat ethylene glycol (EG) and 33mol% ChCl in EG (Ethaline) are shown at 333 K. Similarly to the ChCl/glycerol mixtures, as ChCl is added, the hydroxyl protons experience an upfield shift indicating the chloride anion is interacting with and providing electron density to those protons (lines 109-110). Additionally, dipole rotation time coefficients (Table S11, discussed at lines 283-286) and hydrogen bond dynamics times constants (Table S13, discussed at lines 330-336) are provided for various mol% ChCl in EG.

- Action: Ethaline data added
 - Page 14; 15: BDS data Figure 5d; discussed at lines 257-258
 - Page S13; 6: ^1H NMR data in Figure S13; discussed at lines 109-110.
 - Page S18; 16-17: Dipole rotation time constants in Table S11; discussed at lines 283-286.
 - Page S20; 19: Hydrogen bond dynamics time constants in Table S13; discussed at lines 330-336.

Reviewer Comment

In addition to my previous comments, only two concentrations are studied, 5% and 33% of ChCl. The mixture is an eutectic so it is expected that by a first increase in concentration of ChCl the glass transition temperature decreases, fluidity increases as well as ionic conductivity (all expected behaviours). Likewise I would expect a similar behaviour by approaching the eutectic point starting from 100% ChCl, however, I see that the range of concentration studied does not cover this part of the eutectic and I wonder why this range is not being considered.

Response

We agree with the reviewer that some techniques previously shown only reported studies for a few concentrations of ChCl. We have performed additional experiments and simulations to match the missing gaps in our previous compositions. However, because the focus of the current work is to probe the impact of addition of the hydrogen bond acceptor on the structure and dynamics of the hydrogen bond donor, we are convinced that more insights could be obtained by probing compositions up to the eutectic compositions. The higher concentrations would be more insightful if the properties of the pure ChCl are of interest. However, this is not a trivial experimental task given the premature thermal decomposition at temperatures below the melting point of ChCl. This decomposition complicates spectroscopic assignments of the structural and dynamic features at high ChCl composition for the current systems.

Most techniques employed in this study were able to access the full series of concentrations - 0, 5, 10, 20, and 33mol% ChCl in glycerol. However, for certain techniques, such as QENS and WANS, fewer concentrations were studied due to time constraints and limited beam-times due to travel restrictions in response to the COVID-19 pandemic. We have updated Figure 1 to include the full series of concentrations for ^1H NMR, which follow the trends as expected. Additionally, results of the ethylene glycol based series are provided in the SI.

- Page 6: Figure 1 has been updated to include the entire concentration range.
- Page 14: Figure 5(d) has been added for the entire concentration range of EG/ChCl series.

- Additional data are given in the SI for the entire concentration range of EG/ChCl series.
-

Reviewer Comment

Figure 1, the spectrum reports a resonance labelled as "d", this is however not reported on any of the molecular structures nor explained in the caption. This needs to be revised.

Response

We thank Reviewer 1 for spotting this mistake in our previous manuscript. Figure 1 has been updated accordingly.

- Page 6: Figure 1 has been updated with the correctly labeled scheme.
-

Reviewer Comment

I cannot really see the slight shoulder of peak "i" in Figure 1, bottom.

Response

With the updated Figure 1, at 5 mol% using a shimmering procedure, "i" is observed as a very weak peak because not much ChCl is present. As ChCl concentration is increased, peak "i" becomes more intense.

- Page 6: Figure 1 was updated with new measurements that better show peak "i" in 5 mol%
-

Reviewer Comment

Figure 1, the 5mol% ChCl has much boarder peaks than the 33mol% ChCl. In general, from what I understand from the author's discussion higher viscosity means broader peaks, does that mean that the 5mol% is more viscous than the 33mol%, hence showing broader peaks? I assume this is the case but it should be clarified.

Response

We have revised the discussion at lines 102-104 to clarify that as ChCl concentration is increased, generally, peak broadness decreases. This trend is consistent with viscosities, which also decrease with increasing ChCl concentration.

- Page 5: Lines 102-104 edited to clarify the reviewer's question
-

Reviewer Comment

It is not clear what N_{coord} is, I understand is a coordination number but of what species or group within the molecule? Also, the values are below 1 in general, and with decimal, so I assume this does not refer to number of actual atoms or molecules surrounding the species under investigation?

Response

As the Reviewer correctly pointed out, N_{coord} refers to coordination number, which reflects the number of the coordinating atoms around a reference atom as defined in the caption to table 1 in the original manuscript. Unfortunately, due to reorganization of the contents when we edited the original manuscript, this definition showed up later in the submitted manuscript. This has now been corrected in the revised manuscript. For a given reference atom or molecule at a given time, its N_{coord} is always an integer ($N_{\text{coord}}=0,1,2$, etc.). However, the results reported in the manuscript are ensemble averages over the simulation trajectory, which are usually non-integer and sometimes below 1.

- Page 6: Lines 117 edited to include definition at earliest reference.
-

Reviewer Comment

In the abstract it says “We find that the microscopic heterogeneities induced in glycerol by the addition of the choline chloride lead to new slow, dynamic modes that are strongly correlated to the macroscopic properties of the DES formed”. However, I understand from the text and from Figure 8 that the dynamic becomes faster? This is a bit confusing.

Response

We apologize that we did not provide a clear discussion of the correlation between microscopic heterogeneity and macroscopic transport properties in our previous manuscript. As ChCl is introduced, a new mode or relaxation, slower than the structural relaxation, is observed. However, the rates of all relaxations observed in ChCl/glycerol mixtures are enhanced as ChCl concentration is increased due to increasing molecular mobility. The abstract has been modified to reflect these findings.

- Page 2: Lines 11-14 have been modified to more clearly reflect our findings
-

Reviewer Comment

I find the paper difficult to read in some points. I understand that a lot of techniques and results are included but the way these are presented makes it not easy to follow. The paper would benefit by breaking it down into different sub-sections discuss the results of each technique or group of techniques and conclude highlighting the main

finding for each technique.

Response

This is a great suggestion. Subsections have been added at lines 94, 197, and 342 to compartmentalize the many different techniques into subgroups.

- Page 5: Subheading at line 94 "Local Structure and Microscopic Interactions" added
 - Page 11: Subheading at line 197 "Rotational and Translational Dynamics" added
 - Page 19: Subheading at line 342 "Macroscopic Transport Properties and Emerging Picture" added
-

Response To Reviewer #2

Overall Comments

The manuscript is well written and clear. It provides strong evidence for its conclusions. The methodology is appropriate and impressive, considering the different techniques used and the time and length scale they cover. The quality of the data and quality of presentation are good (some specific details for improvement are provided next). The topic fits well into scope and audience of Nature Communication. However, the work needs major revisions before it can be accepted.

Response

We thank the reviewer for the positive and thoughtful comments. The detailed comments from the reviewer have considerably helped with improving the clarity of the revised manuscript. We address the concerns raised below.

Reviewer Comment

Title and abstract: Although a wide range of complementary techniques have been used in this work, it is just focused in one model DES (Glyceline). The title is too generic, it should be revisited to accordingly reflect the work presented in the manuscript, specifying the effect of concentration and the DES under study. Also, the sentence “These results provide a unified framework for rationalizing the key features of DESs including a decreased glass transition temperature and a corresponding increase in dc ionic conductivity, fluidity, diffusivity, and the mean rates of orientational dynamics” sounds too ambitious. The procedures used may allow doing it, but not the results, as they only apply to glyceline.

Response

We have revised the title to “Evolution of Microscopic Heterogeneity and Dynamics in Choline Chloride-based Deep Eutectic Solvents” to reflect the work that was done. Additionally, we have added data of concentration dependent data of Ethaline (1:2 molar ratio of ChCl to ethylene glycol) throughout the manuscript to highlight that conclusions drawn are applicable to more than just Glyceline.

- Action: Ethaline data added
 - Page 14; 15: BDS data Figure 5d; discussed at lines 257-258
 - Page S13; 6: ^1H NMR data in Figure S13; discussed at lines 109-110.
 - Page S18; 16-17: Dipole rotation time constants in Table S11; discussed at lines 283-286.
 - Page S20; 19: Hydrogen bond dynamics time constants in Table S13; discussed at lines 330-336.
-

Reviewer Comment

The authors have studied the effect of adding ChCl at different concentrations, up to the eutectic concentration in glycerol by different techniques; different temperatures were also chosen, depending on the technique to be used. The selected values of the ChCl concentration are quite consistent among the different methods used (although in some cases only one or two concentrations have been considered); however, different techniques have been used at different temperatures (ranging from 215K to 400K) in a less consistent manner than the concentrations. In fact, it is difficult to follow the effect of temperature with the different techniques used, although, for sure, temperature clearly affects both, the dynamics and the heterogeneity. This is an important issue to be clarified by the authors, and some rationale behind the selections and implications should be presented and discussed. Notice that although some experimental techniques will work better at specific temperatures, simulations can be performed at the same temperatures as the experiments.

Response

Most techniques, experimental and computational, are performed at room temperature. The exceptions are BDS, DMS, and AIMD. We study lower temperatures with BDS and DMS to study the effect of the glass transition on dynamics in a broader range of temperatures. For BDS, a room temperature measurement is made at higher frequencies, shown in Figure S14. The same features are observed, suggesting that the mixtures behave the same at low and high temperatures. This has been made more clear in lines 210-214. Additionally, dynamics become slower as temperature is decreased. This has been clarified at lines 246-247.

- Page 11, 13: Lines 210-214 added to clarify the purpose of measuring across a wide temperature range
- Page 15: Lines 246-247 added to clarify the effect of temperature on dynamics
- Page S13: Figure S14 added of room temperature BDS data at 33mol% ChCl in glycerol

Reviewer Comment

Lines 33-35: it is claimed that little is known about the fundamental relationship between their molecular structure and the macroscopic properties of DES. I agree that considerable work needs to be done to further advance in the field; however, this statement is too strong and not completely accurate, as several experimental and modeling works on DES have been published already, providing very valuable insights into this relationship, as highlighted later in the text with some specific properties/methods. The sentence should be revisited.

Response

We have revised this statement in the updated manuscript in lines 32-36.

- Page 3: Lines 32-36 have been modified

Reviewer Comment

Line 50: it should be “HBA” instead of “HBD” when referring to ChCl.

Response

Thank you for spotting this typographical error. This has been fixed in the revised version of the manuscript.

- Page 3: Line 50 edited to fix the typographical error.

Reviewer Comment

Line 74: please, define QENS

Response

QENS is first defined at line 54 discussing previous literature.

- No changes have been made.

Reviewer Comment

Line 95: The ^1H NMR spectra were obtained at 5 and 33 mol% concentration of ChCl in glycerol (at 333K), while simulations were also performed at 0 mol%, i.e., pure glycerol. Why there are not experimental results from pure glycerol presented? CMD results show the largest deviations for the density versus concentration at pure glycerol (Figure S1).

Also, related to Figure 1, the authors forgot to mention what the peak “d” represents.

Response

We agree that it is useful to have ^1H NMR data at the full range of concentrations, so we performed the NMR experiments for the missing concentrations. Figure 1 has been updated to include 0, 5, 10, 20 and 33mol% ChCl in glycerol. Peak “d” is now properly denoted in the updated figure. We also agree that the CMD predicted density for the pure glycerol has a relatively large deviation from experimental value. However, the deviation is only about 2%, which we consider a reasonably good result and did not make the extra effort to make it perfect (which is not a trivial task).

- Page 6: Figure 1 updated to include full concentration range and correctly labeled scheme.

Reviewer Comment

Regarding Classical Molecular Dynamics Simulations (CMD) and their comparison with experimental data: - The first comparison is shown in figures S1 and S2 in the supplementary information Figure S1: Density as a function of mol% Choline Chloride in Glycerol at 300K for CMD and experimental methods. Although deviations are not large, the slope of the experimental density vs. mol% is different than the one from CMD, being the difference larger near the eutectic concentration. How relevant will the deviations in the density be for the conclusions made? Have the authors check any other force field? Figure S2: Viscosity as a function of mol% Choline Chloride in Glycerol at 300K for CMD and experimental methods. The viscosity calculated with CMD follows the same trend as the experimental data; however, the obtained values are very different, being the simulated ones three times smaller. It is also notice that the problem may arise from the viscosity calculated for glycerol (0% ChCl concentration), where the differences are larger.

- One more point related to the CMD: as the comparison with AIMD is only at 400K and 33 mol% ChCl in glycerol, how accurate is the FF used in CMD for densities and viscosities at 400K? (Figures S1 and S2 show only results at 300K)

Response

We agree with the reviewer that predicted density vs. mol% slope is a little bit off from experiments although the largest deviation in absolute values is only about 3%. A considerable effort was made to select a force field that accurately captured both structure and dynamics of the studied systems. It appeared that currently available force fields are often well parameterized primarily for the eutectic point but have difficulty capturing proper dynamic trends with varying choline chloride mol%. We actually evaluated multiple force fields, and the one used in the current work showed promising results for densities (approximately 3% max deviation with a matching trend), structure, viscosity trends, and rotational dynamic trends within the 0 - 33 mol% ChCl area of focus.

We agree with the reviewer that the predicted viscosities deviate from experimental results quite significantly. Considering the fact that the deviation is different at different ChCl% and that at pure glycerol deviation is the largest, we agree with the reviewer that applied glycerol force field definitely contributes largely to the deviation. On the other hand, we admit the force fields for choline and chloride need to be improved as well. It is known that the absolute values of dynamic properties of ionic systems are hard to calculate using classical force fields. The development of a perfect force field for the systems studied in the current work is likely a lengthy work and beyond the interest of the current study. In the current work, instead of absolute values, our focus is the trends in dynamics, which was successfully captured. Therefore we believe the conclusions drawn based on these simulations provided valuable understanding of the studied systems.

The comparison between CMD and AIMD was made to provide another validation of the CMD simulation. Due to the slow dynamics of the simulated system and expensive cost of the simulation, we were able to generate reliable simulations using AIMD only at 400 K. At 400K and 33mol% ChCl, CMD provided a density of 1.07171 g/cm³ which is 4.76% lower than the same extrapolated density used for the AIMD simulations, which is larger than what we would like to see. However, as shown in Figure 2, all the features in the liquid structure were reproduced by CMD very well. Therefore we believe the applied force field can describe the liquid structure reasonably well and focused on the properties at lower temperatures for the rest of the work. Because we are not

interested in the properties of the systems at this elevated temperature of 400K, viscosity was not calculated.

- No changes have been made.

Reviewer Comment

Page 7: Figure 2: Regarding the RDF and CDF: results are presented just for 33 mol% ChCl in glycerol at 400K (Figure 2). Results are very interesting and clear with this representation; can you comment on the differences observed at other ChCl concentrations and lower temperatures (they should be available at least for the CMD simulations*)? I suggest showing a comparison and the corresponding discussion in the revised version of the manuscript, as both, composition and temperature are relevant to the focus of this work.

Also related to Figure 2: Units for the RDF are in Å (2b) while for the CDF (Combined Distribution Functions) are in pm (picometer) (2c). The units between both figures should be consistent, for an easier visual comparison of the results.

*Note: AIMD were done only at one temperature and one ChCl concentration

Response

We have actually calculated all the RDFs and CDFs but there is little change in RDF shape over the mol% range, and CDFs look identical as they show the same hydrogen bonding behavior. On the other hand, we do agree that seeing the change across multiple mol% is important. To that end, we realized RDFs could be misleading due to the different normalization factors for solutions with different compositions. For these reasons, we did not include other RDFs or CDFs except the ones in Figure 2 which are more for the purpose of a comparison between CMD and AIMD. Instead, in Table 1, we summarized the coordination numbers as well as the locations of the peak maxima and minima at different concentrations and temperatures, which we think are more informative. In addition, in the revised manuscript, we also included a full 300K CMD coordination number table with standard deviations in the SI. While they do not show a different trend, they do provide further important and supporting evidence.

The units in Figure 2 was corrected in the revised manuscript.

- Page 7: Figure 2c units were updated from pm to Å.

Reviewer Comment

Lines 144-149: The authors state: "AIMD simulations at 33mol% indicate that the chloride anion does not selectively coordinate with the Hc or Ht hydrogens of glycerol, with Ncoord of 0.62 (Hc) and 1.27 (Ht), respectively, considering the 1:2 ratio of Hc:Ht in glycerol. This behavior is also seen in CMD simulations at 400 K. These results agree with the 1H NMR data at 333 K in Figure 1, which show a uniform shift of the Hc (e) and Ht (a) peaks as ChCl concentration is increased."

The ¹H NMR data was obtained at 333K, while the CMD simulations were performed at 300 K and 400K (and AIMD only at 400K). The authors comment that the simulations at 400K agree with the ¹H NMR data at 333K; however, different results are obtained at 300K. As 300K is closer to 333K than 400K, simulation results at this temperature are expected to better represent the experimental results. Can you further elaborate on this, please? Have you performed any simulations at 333K? it would be very interesting to see the comparison at the same temperature, and also the effect of temperature with more than two values.

Response

We thank the reviewer for pointing this out and we agree that the results at different temperatures are not a fair comparison. In the revised manuscript, a table of coordination numbers calculated based on a CMD simulation at 340K, the closest available temperature to 333 K, is provided in the SI. The N_{coord} show a near linear correlation between 300 K and 400 K. It does seem that for the CMD simulations, at lower temperatures there is a slight preference for Hc that decreases at higher temperatures. However, the relative ratio for Ht:Hc is still close to a 'no preference' association of Cl.

- Page S21: Table S15 was added detailing coordination numbers and their standard deviation at 340K.
- Page 9: Line 157 was updated to reference the newly added SI table.

Reviewer Comment

Page 9, Table 1 includes the Coordination numbers (N_{coord}) for ChCl in glycerol from CMD (300 and 400K, 5 and 33 mol% ChCl) and AIMD (400K, 33mol% ChCl). - What are the error bars or uncertainties associated to the numbers provided in the table? - The authors claim that (lines 155-156) at 300K, as the ChCl "concentration is increased, a reduced coordination of the oxygen atom on choline by glycerol is observed through a decrease in N_{coord} of Oy-Hc and Oy-Ht"; however, a close look at the values provided in the table show similar numbers at the two temperatures and concentrations (depending on the uncertainties associated to these numbers, which are missing). On the contrary, a clear decrease in coordination numbers of Cl-Hc and Cl-Ht is observed as the concentration increases at 300K. Can you comment on this?

Response

This is a great point. A full table for coordination numbers for all ChCl concentrations with standard deviations is now provided in the SI in the revised manuscript. The numbers are rather close, however with the newly included table with all concentrations and standard deviations, there is a trend of Oy-Hc and Oy-Ht coordinations decreasing with increasing ChCl mol%.

- Page S21: Table S14 was added containing coordination numbers and standard deviations for all concentrations.

- Page 9: Caption of Table 1 was updated to reference the newly added SI Table S14.
-

Reviewer Comment

Line 168: The calculated results from CMD simulations at 0, 5, and – mol% ChCl are also provided. The sentence is missing number “33” before “mol%”

Response

The typo was corrected in the revised manuscript.

- Page 10: Line 186 was updated to fix the typo.
-

Reviewer Comment

12. Page 10, Figure 3: the structure factor at 0 and 5 mol% ChCl from CMD are very similar, and, as the authors pointed out, CMD agrees well with WANS, although the peak at 5 mol% is much more pronounced in the experiments than in CMD. In addition, a “shoulder” is observed at low q in the simulations at 33 mol% that does not appear in the experiments; also, there are two peaks between $2-4q$ that only appear in the simulations. By looking at Figure 3b it seems to be related to the strong Ch-Ch interactions. Can you, please, comment on that?

Response

We agree with the reviewer that there are small deviations between the calculated and experimental $S(q)$. As pointed out by the reviewer, these deviations are due to choline-choline contributions. Similar behavior was also observed in our previous study on Ethaline (see doi/10.1021/acs.jpcc.0c04058) and it was found to be caused by the partial deuteration of choline used in the experiments. This information was added to the revised manuscript.

- Page 10-11: Lines 184-188 was added to address this deviation between experimental and simulated structure factors.
-

Reviewer Comment

Figure 4 - The derivative representation of the dielectric loss is presented in Figure 4a for pure glycerol and 5 mol% ChCl, at different temperatures, ranging from 215-245K. All these temperature values are much lower than any of the values considered in the results previously presented in the manuscript. This goes back to my original comment on the different values of temperatures used for different

techniques. - Figure 4b: temperature?

Response

Like stated in the previous comment, low temperature measurements were made to study the effect of the glass transition on dynamics. The changes made in the manuscript to resolve this issue are listed above. Additionally, the data shown in Figure 4b was obtained from a time-temperature superposition. This detail is now specified in the caption.

- Page 12: Figure 4b caption updated.

Figure 5: the fact that the mean rates of orientational dynamics obtained from the fs-TA, BDS, and DMS are all in quantitative agreement is a remarkable result. Please, indicate what is the meaning of the dashed black line in Figure 5b.

Response

The dashed black line is a guide for the eyes to draw attention to the evolution of the sub- α relaxation. This has been updated in the caption.

- Page 14: Figure 5b caption updated.

Line 237: It is stated that: "Figure 5c shows that the sub-relaxation rate (ω_{slow}) is coupled to the ion hopping rate (ω_{ion}), and both are decoupled from ω_{α} . This suggests that the sub-dielectric relaxation has a similar origin to the ion dynamics, and both rates increase in identical increments with increasing ChCl concentration". This conclusion contradicts those of Faraone et al. and Reuter et al. However, a close look at Figure 5c (within the scale presented there) shows that after a certain concentration, ω_{α} seems to be coupled to the ion hopping rate (ω_{ion}), which goes against the argument of the two rates to be decoupled. Is it a problem of the scale shown in the figure? Please, comment on this.

Response

Yes, when put on a log scale shown here in Figure 1. This makes it more obvious that ω_{ion} and ω_{slow} are decoupled from ω_{α} at all concentrations.

- No changes were made in the manuscript.

Page 15 –Why there are just 2 points for choline E-S in Figure 6c?

Figure 1: The ratios of the various characteristic rates obtained from the dielectric spectra plotted versus mol% ChCl, at a constant T_g/T . Lines are guides for the eyes.

Response

This is because we only have QENS measurements, which give us the mean square displacement to calculate the diffusion coefficients from the Einstein-Stokes equation, at 0, 10, and 33mol% ChCl in glycerol.

- No changes were made in the manuscript.

Lines 300-301: it is stated that: “There is a reduction in both τ_f and τ_b in all pair-wise interactions with increasing ChCl save for Oy-Hy interactions that are not altered from the already low values”. This statement is not completely correct, a look at Table 2 shows that, as expected, both τ_f and τ_b increase for Oy-Hy as the ChCl concentration increases, although their values are smaller than any other O-H values. Interestingly, the average values are smaller at the eutectic concentration than at 20 mol%.

Response

We agree with the reviewer that the average values shown in Table 2 do show some differences in the hydrogen bond dynamics time constants. However, if one takes the standard deviations provided in the SI into consideration, the time constants for Oy-Hy interactions at all concentrations overlap or nearly overlap. Therefore we think it is accurate to state that these time constants are comparable. In the revised manuscript, the following sentence was added/modified to make this point clear: ”(when considering the standard deviations provided in S9)”.

- Page 18: lines 325-326 added to clarify the statement made.

Figure 7 b. Caption: "(b) Fluidity, inverse viscosity, plotted versus inverse temperature for 0-33mol% ChCl in glycerol. Measurements above 298 K were done by another group." What do you mean? Are the results from the other group already published (if so, please, add the reference), if not, some more information should be provided (at least in the acknowledgements).

Response

The group who made the high temperature measurements is included in the author list already (results not previously published). This was only pointed out just to clarify that the values mostly agree across regardless of technique and place prepared. The caption has been revised to specify that low temperature measurements were made with a rheometer, and high temperature measurements were made with a viscometer.

- Page 20: Figure 7b caption was updated.

Line 366: "The addition of ChCl to glycerol creates heterogeneities that are dispersed throughout the mixture". These are microscopic heterogeneities, please, specify.

Response

This change has been made in the revised manuscript.

- Page 23: Line 397 has been updated.

Lines 368-379: "As ChCl concentration is increased up to the eutectic composition, the network becomes more heterogeneous, which weakens the HBN of glycerol, and increases the molecular mobility of all components, observed from QENS, PFG-NMR, CMD simulations, and BDS. This results in the characteristic depressed glass transition temperature of Glyceline." Although the results point in this direction, this, by itself, is not the justification of the presence of the eutectic point, but a consequence of weakening the HBN, also observed in aqueous mixtures. Can you, please, further elaborate on this, and also comment on the influence of temperature?

Response

We have revised this discussion and provided a brief explanation to correlate the microscopic heterogeneities and dynamics to macroscopic transport properties.

- Page 23: Lines 398-407 has been updated.
-

Methodology: 421-422 Simulations were run at temperatures between 280 K and 400 K in 20 K increments for a total of seven temperatures. However, only results at 300 and 400K are presented and discussed. It will be of relevance to present at least the results at 333K, as already mentioned.

We agree with the reviewer and as stated in response to earlier comment, the coordination number results at 340 K, the closest available to experimental temperature of 333 K, have now been included in the SI in the revised manuscript.

- Page S21: Table S15 was added detailing coordination numbers and their standard deviation at 340K.
- Page 9: Line 157 was updated to reference the newly added SI table.

Lines 525-528: references are missing.

Response

Updated.

Response To Reviewer #3

Overall Comments

The amount of data collected for this system is commendable, but I am not convinced that all conclusions are supported by the results.

Response

We thank the reviewer for feedback regarding our results and agree that some of our previous conclusions were too general and not fully supported by the results. We have substantially revised the manuscript to address this issue.

Reviewer Comment

For instance, the possibility of specific solvation (solvation largely by one component within a mixture) of Reichardt's dye is not considered.

Response

Reviewer 3 is correct in pointing out that preferential solvation to betaine-30 probes has been observed in past experiments.[1–3] Indeed, apolar probes elicit hydrophobic solute-solvent interactions (Van der Waals type interactions), while polar groups lead to the possible induction of electronic polarizations, ionic interactions and also possible hydrogen bonding. The solvent systems studied here are of highly polar character, with dominant hydrogen-bonding characteristics. Thus, an apolar probe molecule might have led to different results and different relaxation components. **That is why we have chosen betaine-30 as the probe molecule**, which has a pronounced charge transfer character in its ground state. The charge transfer characteristic implies that the molecule is strongly surrounded by the HBD molecules of the DES, as well as the ionic species, as suggested above.

Furthermore, the molecular simulations in this work suggest a strong hydrogen bonding network in the bulk solvent. The similarity in solvent-solvent interactions in the bulk and the solvent-solute interactions in the surrounding of the probe molecule explains the observed quantitative agreement in the mean rates of the dynamics obtained from fs-transient absorption measurements, broadband dielectric spectroscopy, and dynamic molecular simulation data.

Action: We have added a specific sentence about preferential solvation of Betaine-30, justifying the choice of the probe molecule for our fs-transient absorption measurements, on page 32 (lines 605-619) of the revised manuscript.

Reviewer Comment

My main concern is in regards to the sub-alpha dielectric process, which seems to be visible already in pure glycerol, according to figure 5b, lowest curve. According to

the supplementary material figures, that peak position coincides with the condition where real and imaginary part of permittivity are equal in magnitude, which is known to indicate artifacts due to partly blocked electrodes. This sub-alpha peak is the main pillar of the strong heterogeneity claimed for this mixture, but it may not be a feature intrinsic in the eutectic. Also, the data of Faraone has less indication of such a peak, even if analyzed in the same manner. This component of the results requires careful re-consideration prior to publication.

Response

We respectfully disagree with the reviewer's comment that this process is associated with partly blocked electrodes. As evident from the dielectric data (Figure 4), the process occurs above the frequency corresponding to the onset of electrode polarization, characterized by the increase of dielectric loss with decreasing frequency in the sub- α regime. The sub- α process can therefore not be associated with the nature of electrodes. The reviewer correctly stated that the process is not well-resolved from the dielectric data of Faraone et al. This is presumably the case because, electrode polarization (which can be controlled by adjusting sample thickness, electrodes, among others) shows up at much higher frequencies in their data and entirely overlaps with the timescales of the sub- α process. In contrast, electrode polarization is well separated, especially at lower choline chloride concentrations, from the two relaxations in our data. Please see Figure 4.

We agree with the reviewer that there is a faint signature of the sub- α relaxation in the dielectric spectra of neat glycerol. This feature has been previously observed for glycerol and studied in depth by several groups. It was first attributed to artifacts or gaseous bubbles that exist throughout the HBN of glycerol.[4] However, detailed pressure dependent dielectric studies revealed that this process cannot be explained by artifacts such as gas bubbles but rather the presence of some intrinsic dielectric discontinuities in glycerol.[5] This picture is consistent with our observation that the process speeds up with addition of ChCl to glycerol, with choline occupying the "interstitial voids", or discontinuities, as suggested by Faraone *et al.*[6]. The fact that the sub- α peak corresponds to the frequency at which the real and imaginary parts of the dielectric function coincide reinforces our assignment of the process to ion (choline) re-orientations, also consistent with the ionic liquids literature.

It is worth noting that the sub- α is also observed for the ChCl/ethylene glycol mixtures (Figure 5d), proving that it is not limited to glycerol based systems.

We have revised the manuscript to emphasize that clearly describe the origin of the slow sub- α process, which we attribute to ion-rearrangements. In summary, we present data from both experiments and simulations showing that the slower dynamics originate from ion reorientations in the mixtures studied.

Action: We have explicitly discussed the origin of the sub- α relaxation and attribute it to dynamics associated with ion reorientations. See pages 15 (line 260-265) and 22 (lines 390-396,) of the revised manuscript.

Reviewer Summary

Reviewer # 3 - Summary if present here

Response

Your Response

References

- (1) Singh, P.; Pandey, S. *Green Chemistry* **2007**, *9*, 254–261.
- (2) Wu, Y. G.; Tabata, M.; Takamuku, T. *Journal of Solution Chemistry* **2002**, *31*, 381–395.
- (3) Ray, N.; Pramanik, R.; Kumar Das, P.; Bagchi, S. **2001**, *341*, 255–262.
- (4) Richert, R.; Agapov, A.; Sokolov, A. P. *Journal of Chemical Physics* **2011**, *134*, 74502.
- (5) Casalini, R.; Roland, C. M. *Journal of Chemical Physics* **2011**, *135*, 094502.
- (6) Faraone, A.; Wagle, D. V.; Baker, G. A.; Novak, E. C.; Ohl, M.; Reuter, D.; Lunkenheimer, P.; Loidl, A.; Mamontov, E. *Journal of Physical Chemistry B* **2018**, *122*, 1261–1267.

REVIEWER COMMENTS

Reviewer #1 (Remarks to the Author):

The authors have mostly addressed my comments, the addition of additional systems is helpful. I think the paper can be accepted, but before that I have some final minor point for them to consider. I can see that they added further references, including one on aggregation in ionic liquids, which is relevant to their work. I note this is slightly dated and I would suggest to update this adding in the reference section some more recent works, such as ChemPhysChem 19, 1081-1088. This is also the case since Table 1 reports coordination numbers, which are also discussed somehow in the mentioned previously published paper (the paper discusses aggregation number actually but this is related I think), although for ionic liquids, yet I think they would somehow provide a reference for the value reported in this work, which is missing or not clearly stated. Basically, the numbers in Table 1 would need to be discussed within the context of previous results in ionic compounds. I think after this minor correction the paper can be accepted.

Reviewer #2 (Remarks to the Author):

General comments

Addressing the suggestion of all reviewers about the lack of generality of the original work, the authors have expanded their work on glycerol/ChCl based systems by including ethylene glycol/ChCl systems. A revised title and abstract have been provided taking this into account. The authors have also addressed the questions raised by the reviewers and have reorganized the manuscript for an easier follow up.

The addition of another ClCh-based DES implied a great amount of extra work, which certainly added value to this investigation, in all senses. However, the way in which the new experimental and simulation data has been added and discussed in the revised manuscript needs some improvements. Most of the data of the added DES (Ethaline) has been introduced just in the Supplementary Information (difficult to follow, as explained later), with very few exceptions (the interesting discussion of Figure S13 after Figure 1, page 6; Figure 5d, few other sentences and the methodology section). Although this may be due to space limitations, a better integration into the manuscript is required, especially in the discussion section, still solely based on the original glycerol/ChCl work. Discussing the results of Ethaline is needed, especially considering that the dynamics of this DES could be different than Glyceline, given the fact that hydrogen bond between Cl and EG is much stronger than the EG-EG hydrogen bond (as highlighted by the authors and confirmed by ¹H NMR

data (in Figure S13), resulting in a much higher viscosity than the pure HBD, contrarily to what happens for Glyceline (see lines 331-336). As the experimental and modeling work has already been done, no further work needs to be done in this sense, but the discussion of the data is certainly required.

As mentioned in the previous paragraph, the Supplementary Information section is difficult to follow, as figures and tables appear with not specific order, and somehow, mixed up. I suggest reordering and structuring this important information for an easier follow up by the reader. In addition, the current ordering affects how figures and tables from the SI are named in the main text. Please, notice, for instance, that the first figure mentioned in the text from the SI is called Figure S13 (see page 6, line 110). Figures S2 and S3 appear in the same page, line 119, while Figure S1 is not mentioned in the text. The same applies for Table S14 (see caption in Table 1, page 9) and Figure S15 (line 157, page 9), appearing after mentioning only Table S1 and Table S2 in the text.

Minor comments/suggestions

I am satisfied with most of the changes made in the revised manuscript with respect to the original version addressing the issues raised by the referees (I have focused on comments by referees #2 and #3); however, some points still need further clarifications, or the provided clarifications as answers to the reviewer should be added in the text.

For instance, regarding comment 7 from reviewer #2:

7. Regarding Classical Molecular Dynamics Simulations (CMD) and their comparison with experimental data:

- The first comparison is shown in figures S1 and S2 in the supplementary information

Figure S1: Density as a function of mol% Choline Chloride in Glycerol at 300K for CMD and experimental methods. Although deviations are not large, the slope of the experimental density vs. mol% is different than the one from CMD, being the difference larger near the eutectic concentration. How relevant will the deviations in the density be for the conclusions made? Have the authors check any other force field?

Figure S2: Viscosity as a function of mol% Choline Chloride in Glycerol at 300K for CMD and experimental methods. The viscosity calculated with CMD follows the same trend as the experimental data; however, the obtained values are very different, being the simulated ones three times smaller. It is also notice that the problem may arise from the viscosity calculated for glycerol (0% ChCl concentration), where the differences are larger.

- One more point related to the CMD: as the comparison with AIMD is only at 400K and 33 mol% ChCl in glycerol, how accurate is the FF used in CMD for densities and viscosities at 400K? (Figures S1 and S2 show only results at 300K)

The authors have agreed with the comments made by the reviewer, and have properly clarified different aspects in their answer, but no modifications were made in the revised manuscript.

I fully agree with the statement that “Therefore we believe the conclusions drawn based on these simulations provided valuable understanding of the studied systems.” And with the fact that as shown in “...Figure 2, all the features in the liquid structure were reproduced by CMD very well.

Therefore we believe the applied force field can describe the liquid structure reasonably well.” However, I think it is fair to the readers of the journal to understand that large differences between the simulations and the experiments for the viscosity may come from the limitations of the classical force fields used in the CMD; one short sentence on this could be added to the manuscript.

Page 15: Equation 5 is mentioned in the text, but not provided in the manuscript, neither referred to it as appearing in the SI. Also, please, notice that this is the first time an equation is mentioned in the manuscript (i.e Equations 1-4 appear in the SI, but are not referred to in the text). This goes back to my comment on the SI and its relation to the main body in the manuscript.

377-380 “...choline still strongly interacts with chloride at the eutectic composition, but does not significantly interacting with glycerol, shown by Ncoord. Overall, this creates local, structural heterogeneity, which weakens the HBN of glycerol and drastically increases solvent dynamics”

Please, correct the typo (replace "interacting" by "interact" after "significantly").

Response to the Reviews and Decision

Title: Evolution of Microscopic Heterogeneity and Dynamics in Choline Chloride-based Deep Eutectic Solvents

**Manuscript Reference Number:
NCOMMS-20-24384A-Z**

Authors:

Stephanie Spittle
Derrick Poe
Brian Doherty
Charles Kolodziej
Luke Heroux
Md Ashrafal Haque
Henry Squire
Tyler Cosby
Yong Zhang
Carla Fraenza
Sahana Bhattacharyya
Madhu Tyagi
Jing Peng
Ramez A. Elgammal
Thomas Zawodzinski
Mark Tuckerman
Steve Greenbaum
Burcu Gurkan
Clemens Burda
Mark Dadmun
Edward J. Maginn,
Joshua Sangoro

Date: September 30, 2021

Response To Reviewer #1

Overall Comments

The authors have mostly addressed my comments, the addition of additional systems is helpful. I think the paper can be accepted, but before that I have some final minor point for them to consider. I can see that they added further references, including one on aggregation in ionic liquids, which is relevant to their work. I note this is slightly dated and I would suggest to update this adding in the reference section some more recent works, such as ChemPhysChem 19, 1081-1088. This is also the case since Table 1 reports coordination numbers, which are also discussed somehow in the mentioned previously published paper (the paper discusses aggregation number actually but this is related I think), although for ionic liquids, yet I think they would somehow provide a reference for the value reported in this work, which is missing or not clearly stated. Basically, the numbers in Table 1 would need to be discussed within the context of previous results in ionic compounds. I think after this minor correction the paper can be accepted.

Response

We thank Reviewer #1 for bringing this interesting paper to our attention. We have added the newer reference suggested in line 115 of the revised version of the manuscript for its relevance. However, after much thought we did not add detailed discussion of the coordination numbers in the context of ion aggregation because the extent of ionicity of the DESs studied here is still unclear, in contrast to the ionic liquids mentioned by the reviewer. For instance, our results from broadband dielectric spectroscopy suggest that the DESs studied in this work feature strong dipolar characteristics, presumably due to the dominant contributions of the hydrogen bonded networks. We agree with the reviewer that this would be an insightful research topic and should be explored more in depth in a future study.

Actions:

- We have added the suggested reference in line 115 of the revised manuscript.
-

Response To Reviewer #2

Overall Comments

Addressing the suggestion of all reviewers about the lack of generality of the original work, the authors have expanded their work on glycerol/ChCl based systems by including ethylene glycol/ChCl systems. A revised title and abstract have been provided taking this into account. The authors have also addressed the questions raised by the reviewers and have reorganized the manuscript for an easier follow up.

The addition of another ClCh-based DES implied a great amount of extra work, which certainly added value to this investigation, in all senses. However, the way in which the new experimental and simulation data has been added and discussed in the revised manuscript needs some improvements. Most of the data of the added DES (Ethaline) has been introduced just in the Supplementary Information (difficult to follow, as explained later), with very few exceptions (the interesting discussion of Figure S13 after Figure 1, page 6; Figure 5d, few other sentences and the methodology section). Although this may be due to space limitations, a better integration into the manuscript is required, especially in the discussion section, still solely based on the original glycerol/ChCl work. Discussing the results of Ethaline is needed, especially considering that the dynamics of this DES could be different than Glyceline, given the fact that hydrogen bond between Cl and EG is much stronger than the EG-EG hydrogen bond (as highlighted by the authors and confirmed by ^1H NMR data (in Figure S13), resulting in a much higher viscosity than the pure HBD, contrarily to what happens for Glyceline (see lines 331-336). As the experimental and modeling work has already been done, no further work needs to be done in this sense, but the discussion of the data is certainly required.

As mentioned in the previous paragraph, the Supplementary Information section is difficult to follow, as figures and tables appear with not specific order, and somehow, mixed up. I suggest reordering and structuring this important information for an easier follow up by the reader. In addition, the current ordering affects how figures and tables from the SI are named in the main text. Please, notice, for instance, that the first figure mentioned in the text from the SI is called Figure S13 (see page 6, line 110). Figures S2 and S3 appear in the same page, line 119, while Figure S1 is not mentioned in the text. The same applies for Table S14 (see caption in Table 1, page 9) and Figure S15 (line 157, page 9), appearing after mentioning only Table S1 and Table S2 in the text.

Response

We appreciate Reviewer 2's careful review and helpful comments. We agree that the contents of the SI should be in the order that they are mentioned in the main text. We have revised it extensively to improve the clarity.

Action:

- We re-ordered and restructured the Supplementary Information section to align it with the

order in which the ideas are discussed in the main text. The revised version of the Supplementary Information is now easier to read.

- We removed the original Figure S1, which was not mentioned in the main text. The information is, however, still available as a part of the inset of Figure 1 in the revised main manuscript.

Reviewer Comment

For instance, regarding comment 7 from reviewer #2:

7. Regarding Classical Molecular Dynamics Simulations (CMD) and their comparison with experimental data: - The first comparison is shown in figures S1 and S2 in the supplementary information Figure S1: Density as a function of mol% Choline Chloride in Glycerol at 300K for CMD and experimental methods. Although deviations are not large, the slope of the experimental density vs. mol% is different than the one from CMD, being the difference larger near the eutectic concentration. How relevant will the deviations in the density be for the conclusions made? Have the authors check any other force field? Figure S2: Viscosity as a function of mol% Choline Chloride in Glycerol at 300K for CMD and experimental methods. The viscosity calculated with CMD follows the same trend as the experimental data; however, the obtained values are very different, being the simulated ones three times smaller. It is also notice that the problem may arise from the viscosity calculated for glycerol (0% ChCl concentration), where the differences are larger. - One more point related to the CMD: as the comparison with AIMD is only at 400K and 33 mol% ChCl in glycerol, how accurate is the FF used in CMD for densities and viscosities at 400K? (Figures S1 and S2 show only results at 300K)

Response

We thank the reviewer for pointing out our oversight in not including a fair discussion of some of the shortcomings of the force field. We acknowledge that the computed density shows a slightly greater dependence on composition than the experimental data, but the absolute deviation is still no more than 3% at all compositions and is well within range to support the conclusions made in this work.

As for other force fields, we compared our results against the widely used force field of Perkins *et al.*, which performs extremely well at 33 mol% ChCl but is much worse than our force field at matching the composition dependence of density. We were unable to find any other literature force fields that performed better than our force field for this property.

Regarding viscosity, we completely agree that a major contributor to the lower viscosity is the behavior of the glycerol molecule, and that the simulations systematically underestimate the viscosity at all compositions. However, we believe that capturing the viscosity trend with composition is a good indicator that this force field can support the main conclusions of this work. We are working on improving the glycerol force field for future studies.

The computed density at 400K using CMD is 1.0717 g/cm³. The estimated experimental density at this temperature (extrapolated from lower temperature data) is 1.1497 g/cm³. This results in a CMD density that is 6.8% lower than the extrapolated experimental data. This is

slightly higher than the nominal deviation of 5% that we consider to be the upper limit defining good agreement with experiment. Note, however, that there is good overall agreement between the radial distribution functions and coordination numbers obtained by ab initio molecular dynamics and CMD at 400K.

All things considered, we considered the current force field as a useful tool for extracting composition-dependent features, though it does not perfectly capture density trends nor exactly duplicate viscosity. To this end, we have added a clearer explanation of the strengths and weaknesses of the force field to the manuscript.

Action:

- Added/modified the text starting at line 119 to read “by comparing against experimental results. The computed density as a function of composition matched the absolute experimental values quite well, though the simulations predict a slightly greater composition dependence of the density than is observed experimentally (Figure S2). Although the computed viscosity is lower than the experimental viscosity, the trend with composition is captured very well (Figure S3) and should yield reasonable predictions of composition-dependent dynamics.”

Reviewer Comment

The authors have agreed with the comments made by the reviewer, and have properly clarified different aspects in their answer, but no modifications were made in the revised manuscript. I fully agree with the statement that “Therefore we believe the conclusions drawn based on these simulations provided valuable understanding of the studied systems.” And with the fact that as shown in “. . . Figure 2, all the features in the liquid structure were reproduced by CMD very well. Therefore we believe the applied force field can describe the liquid structure reasonably well.” However, I think it is fair to the readers of the journal to understand that large differences between the simulations and the experiments for the viscosity may come from the limitations of the classical force fields used in the CMD; one short sentence on this could be added to the manuscript.

Response

We agree that a fair discussion of the strengths and weaknesses of the force fields employed in this work should be included. This issue is now appropriately addressed by the new text added from line 119, which incorporated changes consistent with our previous comments quoted by the reviewer.

Reviewer Comment

Page 15: Equation 5 is mentioned in the text, but not provided in the manuscript, neither referred to it as appearing in the SI. Also, please, notice that this is the first time an equation is mentioned in the manuscript (i.e Equations 1-4 appear in the SI, but are not referred to in the text). This goes back to my comment on the SI and its relation to the main body in the manuscript.

Response

We sincerely apologize for this omission. Equation 5 is now provided in the Classical Molecular Dynamics Simulations Methods section. This has been explicitly mentioned at line 273.

Action:

- The equations in the SI were reordered to match the order in which they are mentioned in the main text.
 - Equation 5 is explicitly mentioned at line 273 of the revised main manuscript.
 - Equation S2 is now mentioned at line 362.
-

Reviewer Comment

377-380 "...choline still strongly interacts with chloride at the eutectic composition, but does not significantly interacting with glycerol, shown by Ncoord. Overall, this creates local, structural heterogeneity, which weakens the HBN of glycerol and drastically increases solvent dynamics" Please, correct the typo (replace "interacting" by "interact" after "significantly").

Response

Thank you for this catch. This has been fixed at line 382.

Action:

- The typo mentioned was fixed at line 382.
-